# Analysis of the effect of fish oil on wind waves and implications for air-water interaction studies

Alvise Benetazzo[1], Luigi Cavaleri[1], Hongyu Ma[2], Shumin Jiang[2], Filippo Bergamasco[3], Wenzheng Jiang[2], Sheng Chen[2], Fangli Qiao[2]

[1] Istituto di Scienze Marine (ISMAR), Consiglio Nazionale delle Ricerche (CNR), Venice, Italy.
[2] First Institute of Oceanography (FIO), State Oceanic Administration (SOA), Qingdao, P. R. China.
[3] DAIS – Università Ca' Foscari, Venice, Italy.

*Correspondence to*: A. Benetazzo (alvise.benetazzo@ve.ismar.cnr.it)

**Abstract.** Surfactant layers with viscoelastic properties floating on the water surface damp short gravity-capillary waves. Taking advantage of the known virtue of fish oil to still angry seas, a laboratory study has been made to analyze the wind-wave generation and the interaction between wind waves, paddle waves and airflow. This was done in a tank containing a thin fish oil film uniformly spread on the water surface. The research was aimed, on the one hand, at quantifying for the first time the effectiveness of this surfactant in impeding the generation of wind waves, and, on the other, at using the derived conditions to disentangle relevant mechanisms involved in the air-sea interaction. In particular, our main interest concerned the processes acting on the wind stress and on the wave growth. With oil on the water surface, we have found that in the wind-only condition (no paddle waves) the wave field does not grow from the rest condition. This equilibrium is altered by irregular paddle (long) waves, the generation and evolution of short waves (in clean water and with oil) being modified by their interaction with the orbital velocity of the long waves and their effect on the airflow. Paddle waves do grow under the action of wind, how much being similar in clean and oily water conditions, a fact we ascribe to the similar distortion of the wind vertical profile in the two cases. We have also verified that the wind-supported stress on the oily water surface has been able to generate a surface current, whose magnitude turns out to be comparable to the one in clean water. We stress the benefits of experiments with surfactants to explore in detail the physics at, and the exchanges across, the wavy and no-wavy air-water interface.

**Keywords**: Wind-wave growth; Surfactant; Fish Oil; Marangoni forces; Paddle waves; Air-sea interaction.

# 1   Introduction

It is well known that the addition of an almost mono-molecular film (thickness from $10^{-9}$ to $10^{-8}$ m) of surfactant (blend of *surface active agent*) to the water surface diminishes the energy of gravity-capillary waves by altering the surface tension at the water-air interface (Fiscella et al., 1985). For this reason, oily surfactants were used for centuries by seamen to smooth the ocean surface waves, so much that expressions as "to pour oil on troubled water" have acquired a more general meaning. Crucial in this respect is the type of oil, in particular its polarity. Mineral oils, often used for this purpose during the Second World War, are less effective because their molecules tend to group together. On the contrary the polar molecules of fish, and partly also vegetable, oils repel each other. Hence, once poured on water, they tend to distribute rapidly on the available surface reaching a quasi-monomolecular layer (see e.g. Cox et al., 2017).

Known since ancient time, this damping effect was first studied in the 19[th] century by the Italian physicist Carlo Marangoni (Marangoni, 1872), hence the official name of the process. In relatively recent times the first report of the resonance-type Marangoni damping of wave spectra (briefly sketched in Appendix A) comes from Cini et al. (1983) who had noted the effect in the polluted water, although by mineral oils, in the Gulf of Genoa, Italy. However, clear evidence in the open sea of high damping of surface waves in the short-gravity-wave region by mono-molecular slicks was first verified by Ermakov et al. (1985, 1986) during field experiments in the Black Sea.

The Marangoni damping can be effective for surface waves in two possible conditions. The first one is in open ocean, in which an existing wind-forced wave field travels trough a surfactant patch, with the consequent possibility of recognition (lack of return signal by damped short waves) by microwave radars (Feindt, 1985). A direct application is, for instance, the remote detection of oil spills (Fingas and Brown, 2017); spill drift and deformation, and in turn the damping rate of waves, are affected by the properties of the slick (e.g. the elastic properties of the film) and of the entering wave field (Christensen and Terrile, 2009). Direct in situ observations of the surfactant effects on wind waves have been however very limited because of the challenges to operate in stormy conditions. According to the Marangoni mechanism, for oily surfaces the energy dissipation process is quite selective in wavenumbers, but its effects are not, since it spreads, although to a lesser extent, towards longer and shorter waves via nonlinear interactions and modification of the airflow profile. Hühnerfuss et al. (1983) in a slick experiment carried out in the North Sea found that waves with wavelength up to 3 m are significantly decreased in

amplitude when they travel through a 1.5-km-long monomolecular surface film patch. The result is that the wave growth is inhibited and the existing surface field is rapidly smoothed and progressively attenuated as it propagates within a surfactant patch (Ermakov et al., 1986). This change in surface properties resembles the wave attenuation under an ice cover, for which however there are many other processes which also attenuate energy (see e.g. Weber, 1987, and Christensen, 2005, for a parallel between the two fields). The second condition, typical of laboratory experiments (e.g. Hühnerfuss et al., 1981; Mitsuyasu and Honda, 1986), is where the wind blows over a water surface homogeneously covered with a surfactant film since the rest condition. Compared with a clean water environment, from the wind onset, the coupled air-water system is adjusted to a new state, characterized by strong suppression of the generation of wind waves and a change of the wind stress corresponding to the reduction of form drag of the wave field.

The suppression of wind-generated waves due to the oil slick greatly alters the air-sea interaction process (Mitsuyasu and Honda, 1986), but the full mechanism remains poorly understood (Cox et al., 2017). In this respect, with still different opinions on the reason why (see e.g. Kawai, 1979; Phillips, 1957), we know that as soon as the first wind blows, the sea surface is covered by tiny 2-3 cm long wavelets. That process is quickly overtaken as the waves grow by the feedback caused by the wave-induced pressure oscillations in the air, as soon as the airflow vertical profile is modified by the presence of waves. Miles (1957) proposed a wave growth mechanism that accounts for this change. This theory was extended and later applied by Janssen (1991) to wave forecasting. Its validity is questioned for very short waves whose phase speed is as slow as the air friction velocity (Miles, 1993). According to the shear-flow model by Miles, waves with phase speed $c$ grow when the curvature in the vertical wind profile, at the height (called critical height) where the wind speeds equals $c$, is negative. As a result, the wind profile changes because of the continuous transfer of energy to the waves (Janssen, 1982). The growth rate is proportional to this curvature and it has an implicit dependence on the roughness on the wavy water surface (Janssen, 1991). Hence it is expected that any modification of the vertical wind shear (for instance induced by the oil film effect) modifies the momentum transfer from wind to waves.

In this study, we verify with experiments in a wave tank that the peculiarity of fish oil is indeed striking. Previous researches suggested (e.g. Alpers and Hühnerfuss, 1989) a correlation between the intensity and frequency-range of the wave damping and the chemical properties (namely the dilatational modulus) of the surfactant. Therefore our study explores for the first time the effect of fish oil (a polar surfactant with high

dilatational modulus) by laboratory experiments in which the oil effects on the wind shear stress and the wave
growth are assessed in different conditions for short wind-generated and collinear long paddle-generated waves
The latter ones introduce the problem of the interaction between long and short waves, and between long waves
and the wind profile.

5       In the open ocean, the influence of swell on local wind-wave generation is a known fact, albeit a firm

explanation of the underlying mechanism has not been reached. Hwang et al. (2011) discuss how the Tehuano-
wind generated waves, on the Pacific coast of Mexico, are affected by the incoming oceanic opposing swell. With
some similarity, the cases of following or opposing swell seem to differ somehow in their physics. In the
laboratory, the "following swell" case was first studied by Mitsuyasu (1966) and later intensively by Donelan
(1987) who suggested that a swell (in practice paddle waves; also his experiments were done in a wind- and
paddle-wave tank at the University of Miami, USA) induces a detuning of the resonance conditions for non-linear
interactions among wind waves. Later, in the studies by Phillips and Banner (1974) and Donelan et al. (2010), the
suggested explanation was the enhanced wind wave breaking due to the wind and paddle waves interaction. Also,
the influence of paddle waves increases with their steepness. More recently, Chen and Belcher (2000) proposed the
idea that the long wave exerts a drag on the airflow, which reduces the turbulent stress in the airflow that is
available to generate wind waves.
Given these previous experiences, the present study is motivated, on the one hand, to quantify for the first
time the effect of fish oil on the generation of the gravity-capillary wind-wave field. We worked with waves
generated in a wind-flume using clean water and water with surfactant on the surface. Irregular paddle waves
coexisting with wind waves were also tested to investigate the mutual interaction between short waves, long waves,
and airflow. On the other hand, using the suppressed wind waves with surfactants, we aimed at using the
experiments to disentangle and analyze relevant mechanisms of the air-sea interaction; our interest was on the pure
frictional wind stress (which, in clean water, is not separable from the wave stress), the related water surface drift,
and the growth of irregular long waves under the action of the wind, by largely cancelling the short-wave
roughness.
The paper is organized as follows. Section 2 gives a general description of the experimental set-up in the tank,
lists the general plan of experiments with wind and paddle waves in clean water and with oil, and describes the
water elevation and airflow data collected during the tests. The results of the experiments are examined in Section

3, where we show the response of the coupled air-water system to the presence of fish oil slick. The principal results and implications for air-water interaction studies of the present investigation are discussed in Section 4. Finally, the main findings of the study are summarized in Section 5. The manuscript is complemented with two videos recorded during the experiments showing the effect on wind and paddle waves of the surface layer of fish oil.

## 2   Experiment

### 2.1   The experimental setup

The experiments described in this study were performed in a large wind- and paddle-wave facility allowing the generation of winds at velocities comparable with those in open sea (but no extreme conditions). The measurements were carried out in the flume of the First Institute of Oceanography (FIO, Qingdao, P.R. China) illustrated schematically in Figure 1. The tank dimensions are: length 32.5 m, wall-to-wall cross-section 1.0 m, ceiling at 0.8 m above the mean water surface. The water depth is 1.2 m, satisfying the deep-water condition for the wind-driven gravity-capillary, and practically also paddle, waves analyzed in this study. The smallest longitudinal natural frequency of the tank is 0.052 Hz. Side walls are made of clear glass to enable visualization of the wave field. A water pipe parallel to and below the flume allows the continuity between the two ends of the tank.

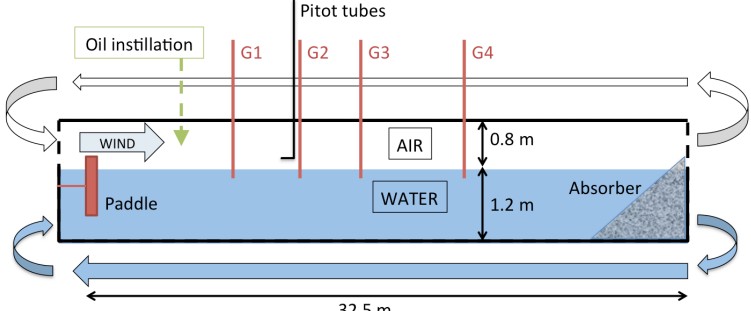

**Figure 1: Schematic longitudinal diagram of the wave flume arrangement at the First Institute of Oceanography flume (Qingdao, P.R. China). For the sake of clarity, horizontal and vertical scales are distorted. The arrows indicate the direction of the flow.**

The wind tunnel is mounted atop the wave tank which is closed with side glass walls and ceiling. The airflow could be driven up to reference speed of 12 m/s. Mechanically-generated (paddle) waves coexisting with wind waves could be generated by a piston-type paddle in the range of periods from 0.5 s to 5.0 s (frequencies from 0.2 Hz to 2.0 Hz) and dissipated at the downwind end of the tank on a sloping beach of fibrous matter. Four calibrated capacitance-type wave gauges (featuring a linear response and accuracy of 0.3 mm) collected

synchronous water surface elevation data for 300 s at 50 Hz at fetches $X = 8.0$ m (probe G1), 12.0 m (G2), 15.5 m (G3), and 20.5 m (G4) along the tank main axis. The fetch $X = 0.0$ m corresponds to the inlet for wind in the water tank. For the statistically steady part of the wave records the variance frequency-spectrum of water surface elevation $z(t)$ was computed by means of the Welch (1967) overlapped segment averaging estimator (eight segments of equal length, 50% overlap, Hamming window).

For characterization of the vertical profile of the along-channel component of the airflow velocity, five Pitot tubes sampling at 1/7 Hz were located at $X = 11.5$ m in the centre of the cross-section and distributed at different heights $h$ above the still water surface, respectively (from tube 1 to 5) at $h = 8.3$ cm, 18.3 cm, 28.3 cm, 38.3 cm, and 48.3 cm. Estimates of the wind stress were derived fitting the Pitot tube observations with a logarithmic law function of $h$. For an aerodynamically rough airflow, the mean value of the along-channel component of the wind velocity $\overline{U}$ in the outer turbulent layer at height $h$ above the boundary is expected to follow a self-similar Karman-Prandtl logarithmic law as a function of height

$$\overline{U}(h) = \frac{u_*}{\kappa} \log\left(\frac{h}{h_0}\right) \quad ( 1 )$$

where the overbar indicates the temporal averaging process, $u_*$ is the friction velocity along the same direction as $\overline{U}$, $\kappa = 0.41$ is the von Kármán's constant. The non-zero parameter $h_0$ has the meaning of the roughness height where $\overline{U}$ appears to go to zero, namely $h_0$ is the virtual origin of the mean velocity profile. In presence of waves, the shape of the wind velocity profile $\overline{U}(h)$ is governed by both turbulent and wave-induced momentum flux, the latter being function of the wind input source term in Eq. ( 11 ). The transport of horizontal momentum due to molecular viscosity is considered negligible, except very near the surface where the vertical motion is suppressed. The total air-side shear stress $\tau_a$ at the boundary of the flow is then approximated as

$$\tau_a = \rho_a u_*^2 \quad ( 2 )$$

with $\rho_a$ the air density.

To account in Eq. ( 1 ) for the non-slip condition, the measured airflow velocity $U_a(h)$ has to be taken relative to along-wind components of the water surface velocity $u_{w0}$ at $h = 0$, i.e. $U(h) = U_a(h) - u_{w0}$. It is thus assumed that the mean water surface drift velocity $u_{w0}$ constitutes the boundary condition at $h = 0$ for the vertical profile of the airflow velocity. Wu (1975) found that at the air-water interface, the wind-induced current is proportional to the friction velocity of the wind, and it is associated to the wind shear, Stokes drift, and momentum

injection during wave breaking events. For non-breaking wavy surfaces, the wind-induced surface current was determined to be around 50% of the airflow friction velocity (Phillips and Banner, 1974 and Wu, 1975). In a clean water wind-wave tank and at steady conditions, the value of $u_{w0}$ can be related to the maximum value of $U_a(h)$. This is not achieved at the largest distances from the water due to the presence of the tank roof. A value $u_{w0}$ around 3.3% of the free-stream maximum wind velocity seems to be a reasonable approximation (Liberzon and Shemer, 2011; Peirson, 1997; Wu, 1975) and was used in this study. The accuracy of the wind stress obtained from the profiles measured by Pitot tubes in a wave tank was examined by Liberzon and Shemer (2011) who found that the values of the wind stress obtained from the measurements by the Pitot tube and those calculated from the Reynolds stresses agree within about 10 % mean difference (using a X-hot-film thermo-anemometer).

A series of videos showing the water surface conditions in clean water and in water with oil complements the available data providing a plain perception of the fish oil effect on the surface waves.

## 2.2 The experiments

Our experiments aimed at analysing the different results, using the different combinations of reference wind speed $U_r$, paddle waves (changing the peak wave period $T_p$ and the significant wave height $H_s$) and oily surface. The data of each experiment were later screened for correct data availability and consistency among the different instruments. This led to exclude several records considered not suitable (independently of the physical results) for the final analysis. This was based on the six experiments listed in Table 1. Two different blower reference speeds were analyzed, namely $U_r$ = 6 m/s and 8 m/s, and one set of irregular paddle waves (JONSWAP spectrum with $T_p$ = 1.0 s and $H_s$ = 6.2 cm; steepness $H_s / L_p$ = 0.04, with $L_p$ the peak wavelength). Note that many of these experiments have been repeated up to four times. All experiments were initiated with no wind and undisturbed water surface.

| Exp. | W06 | W06-O | W06-O-NI | W08 | W08-P | W08-P-O |
|------|-----|-------|----------|-----|-------|---------|
| Wind | 6 m/s | 6 m/s | 6 m/s | 8 m/s | 8 m/s | 8 m/s |
| Paddle | - | - | - | - | JONSWAP Spectrum ($T_p$ = 1.0 s, $H_s$ = 6.2 cm) | JONSWAP Spectrum ($T_p$ = 1.0 s, $H_s$ = 6.2 cm) |
| Oil | - | Instillation | No instillation | - | - | Instillation |

**Table 1: List of experiments performed in the wind- and paddle-wave tank, using clean water (ordinary tap water) and after instilling fish oil. The wind speed is the reference value $U_r$ imposed at the blower. For paddle waves, $T_p$ is the peak period and $H_s$ the significant wave height. Blanks denote not applicable cases.**

Three experiments were made with the fish oil-covered water surface (two with wind waves and one with
wind and paddle waves). The dilational modulus $\varepsilon$ of this type of oil is roughly 0.03 N/m (Foda and Cox, 1980),
hence its resonance frequency given in Eq. ( 10 ) is estimated to be $\omega_{res}$ = 23.7 rad/s, i.e. linear frequency of 3.77
Hz and wavelength of about 11 cm. Moreover, the radial spreading speed of fish oil is around 0.14 m/s, sufficiently
large to keep uniform the oil film that might be broken by the wave action (Cox et al., 2017). Before performing
the experiments with the slick-covered surface, the oil was instilled from the ceiling at fetch $X$ = 4 m, releasing 26
drops with the blower at rest. Estimating each drop of volume about $50 \times 10^{-9}$ m$^3$, the average oil thickness on the
water surface was $4 \times 10^{-8}$ m, namely a few molecular layers. During the experiments W06-O and W08-P-O the oil
film was preserved by continuously instilling onto the water surface oil drops, while the dropping was interrupted
(no instillation) at the onset of the wind start in the experiment W06-O-NI (see Table 1).
A crucial point in wind wave tank measurements concerns the correct reference system for waves generated by
wind. The wave data acquired by the probes in the tank are represented in a fixed (absolute) reference system,
while the response of the wave field to the oil film is intrinsic in the wave dynamics. Therefore the sea surface
elevation energy spectrum $E$ must be mapped in a reference system moving with the wind-generated near-surface
water current. To this end, the wave spectrum must therefore be transformed from absolute $f_a$ to intrinsic
frequencies $f_i$, i.e. those that would have been recorded by a probe moving with the current. Indeed, for waves
propagating over a moving medium, the Doppler effect modifies the observed frequency of each elementary
periodic wave that makes up the random wind field (Lindgren et al., 1999). This effect can be particularly large for
short waves at sea (hence modifying the slope of the high-frequency spectrum tail; Benetazzo et al., 2018), and so
in a wave tank where waves are generally short whilst the current speed can be a non-negligible fraction of the
wave phase speed.
For harmonic waves in the limit of small wave steepness and neglecting the modulation of short waves by long
waves (Longuet-Higgins and Stewart, 1960), the relation between $f_a$ and $f_i$ is given by (Stewart and Joy, 1974):

$$f_a - f_i - [ku_w \cos(\theta - \theta_U)]/(2\pi) = 0 \qquad ( 3 )$$

where $u_w$ is an appropriate water velocity vector of direction $\theta_U$, and $\theta$ the wave direction (Kirby and Chen, 1989;
Stewart and Joy, 1974). At the leading order, it is assumed that the dispersion relationship of the gravity-capillary
wave theory provides a unique relationship between the frequency $f_i$ and the wavenumber $k$ as follows:

$$2\pi f_i = \sqrt{gk + \frac{T}{\rho}k^3} \qquad (4)$$

with $T$ the water surface tension, $\rho$ the water density, and $g$ the gravity acceleration. In accordance to Eq. ( 3 ), the spectral representation in absolute frequencies experiences a shifting of the energy distribution (see Figure 5 below). In our case, we consider that short waves in the tank mostly feel the surface current drift but we neglect the Doppler shift associated with the orbital motion of long waves. That drift can be estimated by the wind speed, and in the tank assumed aligned with the waves (namely, $\theta = \theta_U$). Hence, the frequency spectrum in intrinsic coordinates can be derived as

$$E(f_i) = E(f_a)J_{ai} \qquad (5)$$

where $J_{ai} = |df_a / df_i|$ is the Jacobian of the transformation, which in the limit of deep water can be written explicitly as

$$J_{ai} = |1 + 2u_w \sqrt{gk + \frac{T}{\rho}k^3} \Big/ \left(g + 3\frac{T}{\rho}k^2\right)| \qquad (6)$$

## 3   Results

### 3.1   Wind waves without and with oil

We begin the examination of the change of gravity-capillary wave properties caused by the fish oil film by analysing the effects on the water elevation $z$ and wind speed profile $U_a(h)$ during the experiments W06 ($U_r = 6$ m/s and clean water) and W06-O (the same as W06, but with oil slick). For the latter, the wind wave field attenuation due to the Marangoni forces is readily visible in Figure 2 that shows two pictures of the water surface without (left panel) and with (right panel) oil instillation. After the oil film is spread on the water, the surface is largely smooth, with only tiny elevation oscillations (1 mm at most; see Figure 4) and there appears to be no organized wave motion (see also the video available as supplementary material SM1: https://doi.org/10.5281/zenodo.1434262). It is obvious that the presence of an extremely thin, practically mono-molecular, layer of oil on the surface greatly alters the air-sea interface properties. We analyze the situation first from the air, then from the water, point of view.

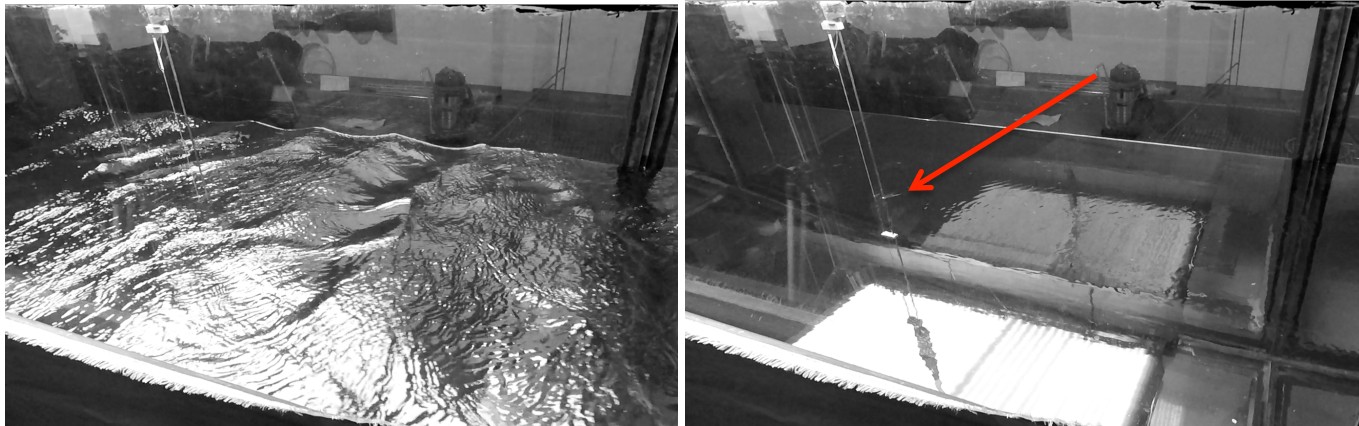

**Figure 2: Two photographs showing the water surface condition at fetch of about 20 m taken without (left panel) and with oil instillation (right panel). On both cases the water surface was forced with a reference wind speed $U_r$ = 6 m/s (blowing from left to right on the pictures). The wave probe G4 is visible on the left-hand side of both pictures. The red arrow in the right panel points to the oscillating flow (vortex shedding) past the probe. See also the video available as supplementary material SM1.**

### 3.1.1 Airflow characterization

We begin evaluating the air-side stress due to the wind drag on the water surface. In the experiment W06 the maximum value of $\overline{U}_a(h)$ of 6.06 m/s was found, attained at the third Pitot tube, namely at $h$ = 28.3 cm from the still water surface. Hence, accounting for a surface water speed $u_{w0}$ = 6.06x0.033 ≈ 0.2 m/s, a logarithmic curve was fitted (the mean absolute error of the fitting is 0.01 m/s) to the average profile $\overline{U}(h) = \overline{U}_a(h)$ - $u_{w0}$ and the two parameters $u_*$ = 0.29 m/s and $h_0$ = 0.1 mm were estimated accordingly. For the lowest Pitot tube ($h$ = 8.3 cm), the dimensionless height $h_* = gh/u_*^2$ is 9.7, implying that from there upward wind data were collected in a region where turbulent stresses are expected to dominate over wave-induced stresses (Janssen and Bidlot, 2018). The typical viscous sublayer thickness approximated as $11.6\nu/u_*$ measured 0.6 mm, and the atmospheric boundary layer may be characterized as aerodynamically smooth. The wind shear stress $\tau_a$ based on the measurement in this layer equals 0.10 N/m$^2$. The equivalent wind speed at height $h$ = 10 m, extrapolated from Eq. ( 1 ), is $\overline{U}_{10}$ = 8.6 m/s, which is in satisfactory agreement with the typical relation between $u_*$ and $\overline{U}_{10}$ found in a wind-wave tank filled with clean water (Liberzon and Shemer, 2011; Mitsuyasu and Honda, 1986). The neutral drag coefficient measured at 10-m height and defined as $C_D = \left(u_*/\overline{U}_{10}\right)^2$ is 1.1x10$^{-3}$. The values of $u_*$ and $h_0$ are in agreement with those obtained in the experimental results of Liberzon and Shemer (2011), and those estimated by the bulk parameterization of air–sea turbulent fluxes provided by the Coupled Ocean–Atmosphere Response Experiment

(COARE) algorithm (Fairall et al., 2003) that gives $\tau_a = 0.11$ N/m$^2$, using as input the profile $\overline{U}(h)$ determined in
our experiments.

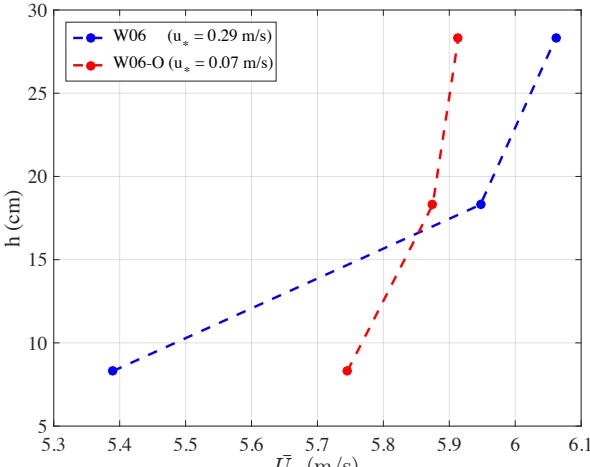

**Figure 3: Vertical profile of the wind velocity component $\overline{U}_a$ measured over the water surface at fetch $X = 11.5$ m. Reference wind**
**speed $U_r = 6$ m/s. Clean water conditions (blue) and oil-covered surface (red). In the legend, the value of the airflow friction velocity**
**$u_*$ is shown within brackets. Only the value recorded at the three lowest Pitot tubes are shown.**

7       The smoothing of the water surface in presence of oil reflects on the airflow, which differs from that over

clean water and has a smaller vertical gradient $d\overline{U}_a/dh$ of the wind speed (Figure 3). Because of the reduction of
the resistance on the water surface, the wind speed strengthens over the film-covered surface, but the effect is
limited to the lowest part of the turbulent airflow. Similar behaviour was observed in the wind-wave tank
experiments by Mitsuyasu and Honda (1986). For continuity reasons, i.e. for the practically constant airflow
discharge in the tank (the difference of the discharges measured by the Pitot tubes is smaller than 1%), a less steep
wind profile implies a lower velocity with respect to the clean water case in the central line of the flow (with oil the
maximum value of $\overline{U}_a(h)$ was 5.91 m/s).

15       In the case of the oil film-covered surface, a problem arises, i.e. if the 3.3%-rule still holds to determine the

surface current drift from the wind speed. The problem stems from the fact that for clean surfaces the momentum
flux to the water column (i.e. for the generation of current) is the sum of the flux transferred across the air-sea
interface not used to generate waves and the momentum flux transferred by wave breaking. In terms of spectral
quantities the stress to the water column $\tau_w$ can be computed as

$$\tau_w = \tau_a - \rho_w g \int_0^{2\pi} \int_0^{k_{max}} \frac{k}{\omega} (S_{in} + S_{nl} + S_{di}) \, dk d\theta \qquad (7)$$

where we have omitted the direction of the flux that we assume aligned with the flume main axis. On the right side
of Eq. ( 7 ), the $S$ terms represent the net effect of sources and sinks for the wave energy spectrum (see Appendix
A). In the high-frequency equilibrium range, the momentum coming from the wind and nonlinear interactions is
dissipated, and is therefore directly transferred to the water column.

5          The balance in Eq. ( 7 ) is plainly altered for an oil-covered surface, as it is visible in the right panel of

Figure 2, for which we assume $S_{in}+S_{nl}+S_{di} = 0$. In this case, the wave-induced transport is practically null, and
the total current drift is supported only by the stress $\tau_a$ exerted by wind at the air-sea interface. In this respect, from
visual inspection of the supporting movies acquired during the experiments, we did observe the presence of a water
surface drift and a high-frequency oscillating flow downstream from the probes' beams (the vortex shedding at G4
is pointed by the red arrow in the right panel of Figure 2; see also the video available as supplementary material
SM1). The latter implies the presence of a near-surface drift impacting the probes. No adequate instrumentation
(e.g. Particle Image Velocimetry; see e.g. Adrian, 1991) had been designed in advance to obtain a representation of
the fluid flow close to the wavy air-water interface.

14        However, the availability of video-camera images allowed two independent estimates of near-surface water

drift. The first one was possible tracking the motion of tiny bubbles moving on the water surface along the tank and
clearly visible in the 1920x1080 pixel images captured at 60 Hz by a video-camera placed outside the tank, close to
the probe G4. A detailed description of the procedure is given in Appendix B. The distribution of the so-defined
surface speed has a mean value of 26 cm/s and standard deviation of 11 cm/s. Despite the relatively large
variability, such observations clearly show that, even if the stress on the water surface is largely reduced by the oil
film, a surface current is still present whose order of magnitude is comparable with what one expects in clean
water. A possible objection to this approach is that the bubble motion could be due, partly at least, to the wind drag.
However, this estimate, albeit with some approximation, is supported by the second indirect estimate. On the right
panel of Figure 2 the wave probe across the surface is clearly visible, and there is a "wake" behind it. Indeed, as we
will soon discuss, the wave spectra show an isolated peak around 10 Hz that we interpret as due to the vortex
shedding caused by the surface current flowing around the probe support (diameter $d = 4$ mm). Use of the related
vortex shedding frequency $f \approx 0.21 u_{w0,oil} / d$ (see later in subsection 3.1.2) suggests $u_{w0,oil} = 20$ cm/s that is close
(actually less than) the estimate using bubbles, which probably were partially also drifted by wind.

28        With this information, we have found that the wind stress for oil-covered smooth surface is $\tau_{a,oil} = 0.005$

N/m$^2$, approximately 5% of $\tau_a$, and the drag coefficient undergoes a decrease of one order of magnitude. The
extrapolated 10-m height wind speed $\overline{U}_{10,\text{oil}}$ is 6.3 m/s, smaller than $\overline{U}_{10}$, in contrast to the field observations of
Ermakov et al. (1986), most likely because our observations were collected in a tank with an upper roof. The
roughness height was determined to be $h_0 < 10^{-6}$ m, implying that the air boundary layer for the oil-covered water
shows properties of a hydrodynamically smooth flow. This result complements those of Mitsuyasu and Honda
(1986) and Mitsuyasu (2015), who observed that for low wind speeds (few metres per second) and short fetches the
water surface aerodynamic properties are similar in clean water and in water with surfactant. Indeed, in those
studies the surfactant only partially suppressed the wind-wave field, while, on the contrary, the use of fish oil in our
experiments cancels the wind-wave generation process such that the water surface is felt as smooth by the airflow.
### 3.1.2    Wave field characterization
In presence of the viscoelastic oily film, the gravity-capillary wave damping is quantified by analysing the time
records $z(t)$ of the sea surface elevation field at different fetches. In this respect, Figure 4 (left panel) gives a clear
idea of the Marangoni damping effect, which can be quantified by noting that the standard deviation $\sigma$ of $z(t)$
shrinks by one order of magnitude. However, the process involves more than a decrease of the vertical oscillations,
as it is the whole spatio-temporal distribution of the surface elevations that is abruptly changed (right panel of
Figure 4). Indeed, whereas in clean water, in active wave generation, the histogram of $z$ (high-pass filtered above 1
Hz; see discussion below) has a positive skewness coefficient, as it is expected for wind-waves (Longuet-Higgins,
1963), in presence of oil the empirical histogram is quasi-symmetric around the mean (the skewness coefficient is -
0.03, very close to zero). This implies that for slick-covered surfaces the generation and evolution of gravity-
capillary waves are governed by different balance and process, which are dominated by the reduced wind input and
the Marangoni energy sink, which lead to a quasi-Gaussian surface elevation field at all scales.

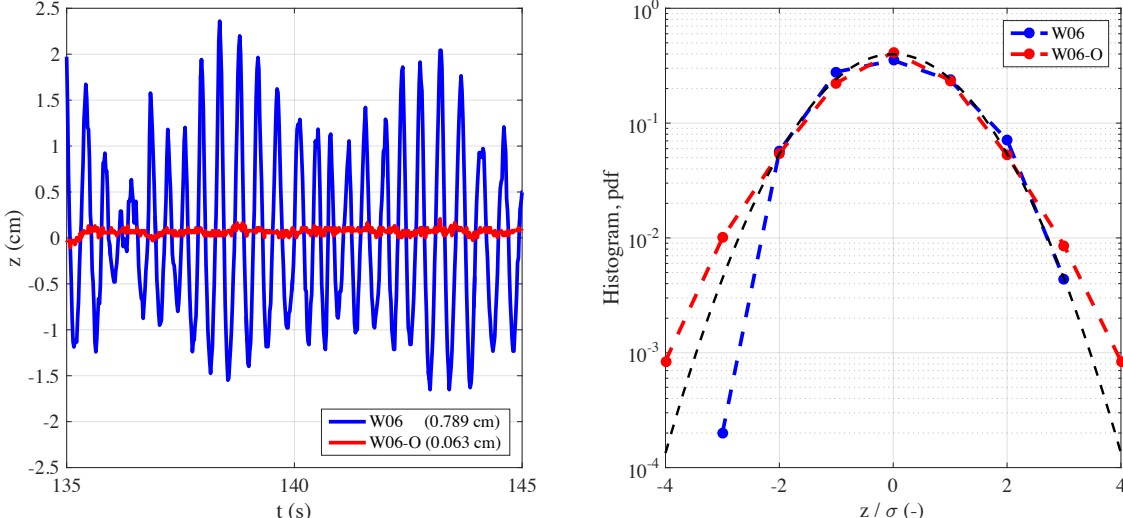

**Figure 4: Sea surface elevation for clean (blue) and slick-covered (red) surface. (left) Excerpt of the single-point record $z(t)$ at $X =$ 20.5 m (probe G4) for the two conditions with clean water (experiment W06) and after the oil instillation (W06-O). In the legend, the value of the standard deviation $\sigma$ of $z(t)$ is shown within brackets. (right) Histogram of the normalized elevations, high-pass filtered above 1 Hz. The black dashed line shows the Gaussian probability density function (pdf).**

The most general and quantified view of the effect of oil is provided by the energy spectra $E(f_a)$ and $E(f_i)$ of water elevations (respectively absolute and intrinsic frequency). These are shown in Figure 5, for experiments W06 (left panel) and W06-O (right panel). For a more direct comparison the G4 oil spectrum is reproduced in the no-oil diagram (dashed blue line). The spectrum $E(f_i)$ was computed using the Jacobian transformation described in section 2 and assuming all wavenumbers are shifted by the surface current $u_{w0} = 0.2$ m/s. Note in the "intrinsic" spectrum the expected shift towards lower frequencies, more evident in the right side of the spectrum where higher frequencies move with a lower speed with respect to the current. From now on we only deal with the intrinsic-frequency quantities. Starting with clean water conditions (left panels), the variation of the wave spectra with fetch is characterized by the expected downshift and overshoot of the peak of the spectrum. The total wave energy increases with fetch: the significant wave height $H_s$ grows from 1.21 cm at the shortest fetch ($X = 8$ m) to 3.16 cm at $X = 20.5$ m. It is remarkable that for the slick-covered surface, there is no evidence of wave growth with fetch (right panels).

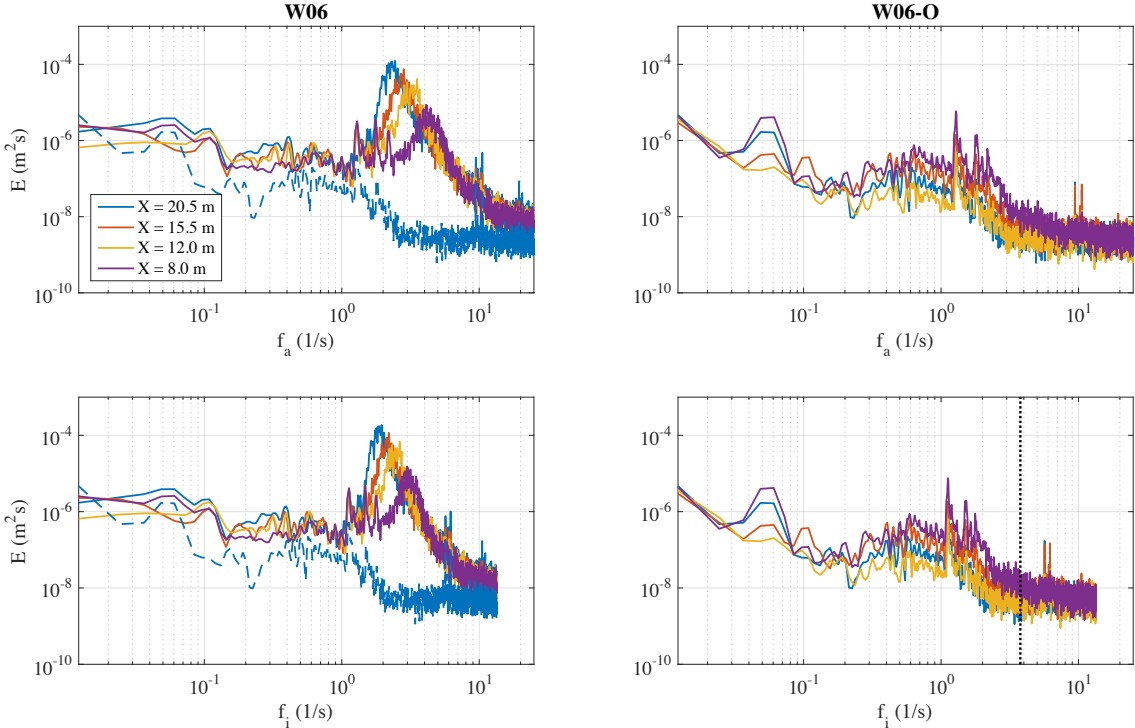

**Figure 5: Wave energy spectra $E(f_a)$ and $E(f_i)$ of the water surface elevation $z(t)$ at different fetches. Reference wind speed $U_r = 6$ m/s.**
**(left) Clean water experiment W06. (right) Water with oil experiment W06-O (the spectrum at the fetch $X = 20.5$ m is replicated with**
**a dashed blue line on the left panels). The vertical grey dotted line on the bottom-right panel shows the fish oil resonance frequency**
**$f_{res} = \omega_{res}/(2\pi) = 3.77$ Hz.**
Focusing for the time being on the comparison among the oil and no-oil cases, the differences are obviously
macroscopic, but it is worthwhile to analyse them for different frequency ranges. For low frequencies, say below 1
Hz, there is clearly some energy also in the oily spectra. Note the peaks around 0.05 Hz at the G1 and G4 spectra,
reduced at G2 and G3. Remembering (see Section 2) the longitudinal natural frequency of the tank, we interpret
these as "seiches" of the wave tank, obviously more visible the further the gauges are from the centre of the tank. In
the clean water case more distributed oscillations exist, that we associate to a more active action of a possibly
irregular wind flow. The most interesting range is of course between 1 Hz and 4 Hz (close to the fish oil resonance
frequency). Here the effect of oil is macroscopic, with oil-case energy several orders of magnitude smaller than
without oil. Finally, still for the oil spectra, no wave signal is visible above 3-4 Hz where we expect the maximum
damping of surface waves due to the oily surfactants (Alpers and Hühnerfuss, 1989).
A more direct comparison between the W06 and W06-O spectra is obtained showing in Figure 6 for each
frequency the ratio of the respective spectral energies. If we represent the frequency/fetch-dependent damping
coefficient as the ratio $D(f_i, X) = E_o / E_c$ between the variance density spectrum of the water surface elevation with
oil slick ($E_o$) and in clean water ($E_c$), we then find $D$ values as small as $10^{-4}$ at the longer fetches. Of course the
maximum differences are at the peak frequency of the no-oil spectra, the respective frequency and ratio decreasing
with fetch while the no-oil energy increases.

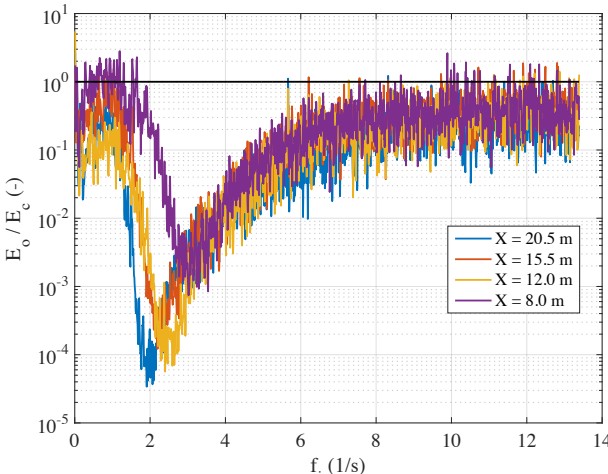

**Figure 6: Damping of the wave energy for oil-covered water surface. The damping coefficient is evaluated as the ratio $D = E_o / E_c$**
**between the variance density spectrum of the water surface elevation with oil slick ($E_o$) and in clean water ($E_c$). The solid black thin**
**line shows the level $E_o = E_c$.**

8       To interpret the data from the experiments we analyze how wave energy depends on fetch. In Figure 7 we

show how the corresponding surface elevation (high-pass filtered above 1 Hz) variance $\sigma^2 = \langle (z(t) - \langle z(t) \rangle)^2 \rangle$
(the angle brackets $\langle \; \rangle$ denote the ensemble average) varies for the clean water and oil cases. Granted the different
orders of magnitude, it is macroscopic that, while the wave energy in clean water grows with fetch, the opposite is
true (or is suggested to be) with oil. To quantify better the fetch dependence we have fitted a power law

$$\sigma^2 = \alpha X^\beta \qquad (\,8\,)$$

to the water surface variance $\sigma^2$ versus fetch $X$. The best fit parameters $\alpha$ and $\beta$ are tabulated, respectively, in the
legend of Figure 7 (with $\sigma^2$ in cm$^2$ and $X$ in m).

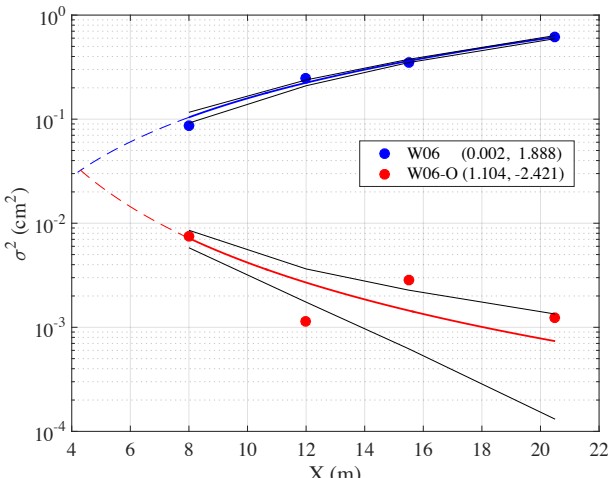

**Figure 7: Spatial evolution of the water surface elevation variance $\sigma^2$ (filled markers) in clean water (W06) and with fish oil slick (W06-O). The solid blue and red lines are the power law-type function $\sigma^2 = \alpha X^\beta$ fitting the observations (in the legend, the coefficient $\alpha$ and $\beta$, respectively, are tabulated), while the dashed lines show the laws extended outside the observed data interval. The root-mean-square error for the fitted function is 0.027 cm$^2$ and 0.002 cm$^2$ for W06 and W06-O, respectively. The black lines show the prediction bounds (confidence level of 50%) for the fitted curves.**

Prolonged in the figure backwards out of the experimental range, the two fitted laws intersect each other around $X = 4$ m. This is the fetch at which oil was introduced into the tank. Note that in the W06-O experiment oil was continuously instilled during the experiment. This was because the wind, acting on the surface oil and creating, as we have seen, a current, tends to push it along the tank faster than the oil tends to distribute uniformly on the surface (with radial speed around 14 cm/s). Indeed this is smaller than the surface speed derived in the previous sub-section for the oil-case. While the continuous, although very limited, instillation of oil during W06-O ensured the presence of oil film from the instillation point onwards, the wind, acting from fetch $X = 0$ m, was pushing the oil away from the first four-metre zone where therefore waves could be generated, hence equal, in both the oil and the no-oil cases. Therefore the "oil" energy we see in Figure 7 at eight-metre fetch is the remnant of the one previously generated and already partially dissipated between the 4 and 8 m fetches due to the acting Marangoni forces. This explains why the highest energy in the oily spectra (right panel) is in the first spectrum, i.e. the shortest fetch.

The shift along the tank of the surface oil film due to the wind drag is also well illustrated by the results of the experiment W06-O-NI, i.e. when, starting with a layer of oil well distributed on the water surface in the tank, we did not further instillate oil during the action of the wind (reference wind speed $U_r = 6$ m/s). The resulting records at $X = 15.5$ m (G3) and 20.5 m (G4) are shown in Figure 8. It is obvious that around 262 s the effect of oil

is beginning to vanish at G3, followed 15-20 s later by a similar result at G4. Note that this does not mean the
whole oil was pushed past G3 at 262 s. Were this the case we should see in the record the already generated waves
(up to position G3). Rather, the oil edge is getting close enough to let G3 feel the consequences, which are different
from those at G4: in the range [290 s, 300 s] $H_s$ grows from 1.56 cm (G3) to 2.34 cm (G4), conveying the fact that
a longer fetch was progressively made clean by the near-surface water drift. The progressively increasing space
free of oil is also manifest in the record of each probe, where the basic wave period tends to increase with time.

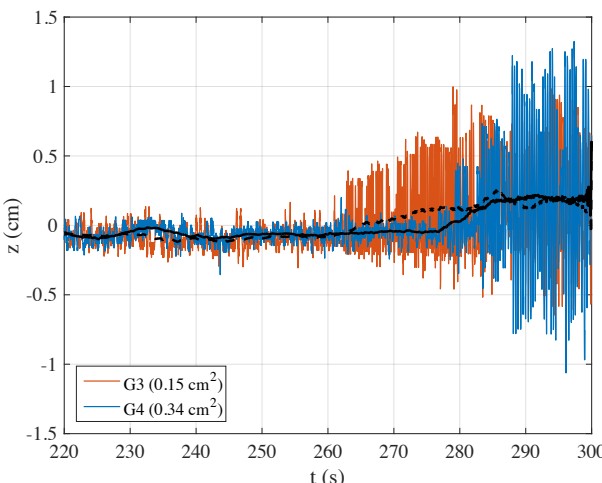

**Figure 8: Sea surface elevation at probes G3 ($X$ = 15.5 m) and G4 ($X$ = 20.5 m) for an initially slick-covered surface without oil**
**instillation (experiment W06-O-NI). The dashed and solid black lines show the smoothed elevations (moving average of size 5 s) at**
**G3 and G4, respectively. In the legend, the variance of $z(t)$ within the range of 290 s ≤ $t$ ≤ 300 s is reported.**

11       In the experiment W06, spectra at the longer fetches show two highly energetic and very close peaks

around the frequency $f_a \approx 10$ Hz (probes G3 and G4, see Figure 5). As mentioned in section 3.1.1, our interpretation
is that they are originated by the vortex-induced vibrations at the cylindrical holding beams of the probes. Indeed,
the frequency $f_v$ at which vortex shedding takes place is related to the Strouhal number by the following equation:

$$St = \frac{f_v d}{u} \qquad ( 9 )$$

where $St$ is the dimensionless Strouhal number, $f_v$ is the vortex shedding frequency, $d$ is the diameter of the body,
and $u$ is the flow velocity. The Strouhal number depends on the Reynolds number, but a value of 0.21 is commonly
adopted (Steinman, 1946). Adopting $d$ = 0.4 cm (the diameter of the probe's holding beam) and $u$ = 20 cm/s, the
vortex frequency is $f_v$ = 10.5 Hz, consistent with the experimental evidence. Elaborating this point further, it is
worth noting that similar spectral peaks (as energy and frequency) have been found also during the W06-O
experiment. In our interpretation, this evidence supports the fact that the water surface drift was generated by the
wind friction also in presence of oil, and that its magnitude is consistent with the one expected in clean water
conditions.

## 3.2 Wind and paddle waves without and with oil

The second series of experiments was done adding mechanically-generated paddle waves to the wind-generated
ones, both in clean water and in water with fish oil. We had two specific purposes. The first one was to explore the
influence of pre-existing relatively long waves (the paddle generated ones) on the local generation of wind waves.
The second purpose was how this interference was modified by the presence of fish oil. This second set of
experiments (namely W08-P and W08-P-O) was done with 8 m/s reference wind speed. A 6 m/s reference speed
would have allowed a more direct comparison with the results obtained without paddle waves (previous section).
At the same time, a higher wind speed was useful to better highlight the interaction with the paddle waves.

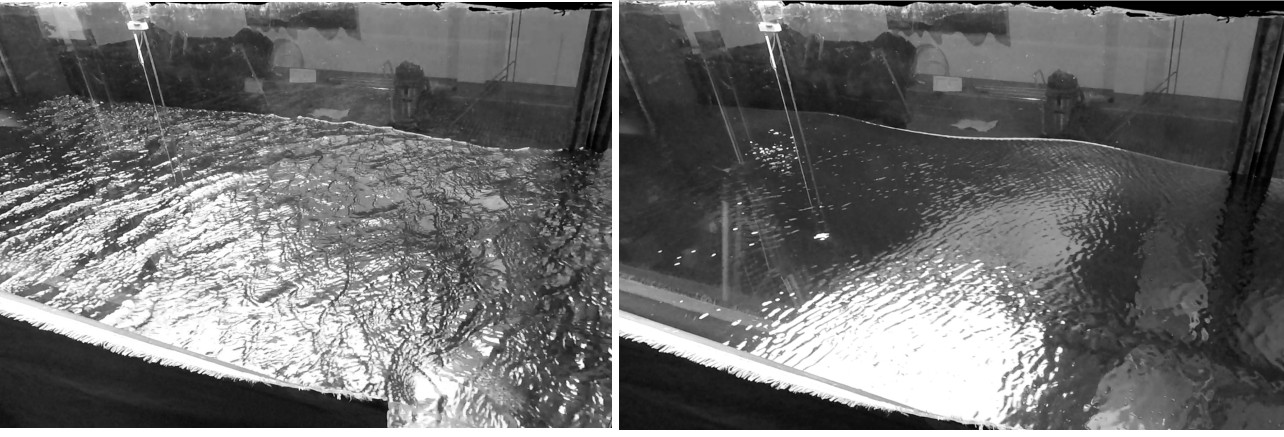

**Figure 9: Two photographs of the water surface condition at fetch of about 20 m taken without (left panel, experiment W08-P) and**
**with oil instillation (right panel, experiment W08-P-O) onto wind-waves (wind blowing from left to right of the pictures at the**
**reference wind speed $U_r$ = 8 m/s) and irregular paddle-waves. See also the videos available as supplementary material SM2.**
In the tank paddle waves were generated as a JONSWAP spectrum with nominal 6.2-cm significant wave
height and 1.0-s peak period. On top of this wind waves were generated by the wind. The resulting surface wave
field is shown in Figure 9 (left panel), the picture being taken, as Figure 2, close to fetch $X$ = 20.5 m. Starting with
a qualitative perception from the image (see also the videos available as supplementary material SM2), we derive,
as expected from what is reported in the literature, that wind waves grow substantially less than expected in a pure
wind sea (Wu, 1977). For a full comparison we also ran the W08 experiment, with wind blowing at $U_r$ = 8 m/s
without paddle waves and in clean water. The comparison between Figure 2 and Figure 9 is even more striking
considering the larger wind speed in Figure 9. Clearly the presence of the paddle waves has an effect. This is a
matter of practical relevance for the cases when in the ocean fresh new waves are generated superimposed on a pre-
existing swell (in this case propagating along the same direction). A more quantified comparison of W08-P versus
W08 (i.e. with versus without paddle, respectively) is provided by the wave spectra shown in Figure 10. We see
that the introduction of the irregular paddle waves cancels the wind wave peak of W08 at about 1.6 Hz. However,
the tails of the two spectra somehow converge above 2.2 Hz. As we will soon see, with paddle waves the
conditions did not allow the visual measurement of current.

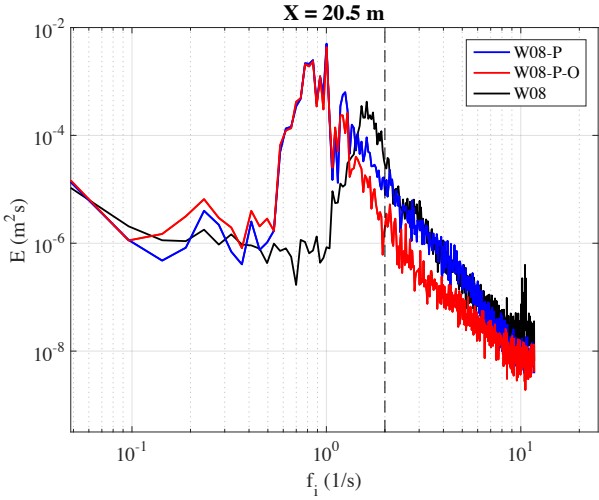

**Figure 10: Variance density spectrum $E(f_i)$ of the water surface elevation $z(t)$ at fetch $X = 20.5$ m (wave probe G4) in presence of co-**
**existing wind and paddle waves in clean water (W08-P), wind and paddle waves in water with oil (W08-P-O), and wind waves only in**
**clean water (W08). The dashed grey vertical line shows the maximum frequency (2 Hz) produced by the paddle.**

12          We analyze how the overall effect varies with fetch (both without and with oil). The related spectra are in

depicted in Figure 11. Contrarily to the pure windy cases (experiments W06 and W08), there appear to be no
evident dependence on fetch of the wave energy. Our interpretation is the following. On the one hand the
disappearance of the wind sea energy peak in presence of long waves implies that the wind wave peak does not
develop with fetch. On the other hand, in water with oil, at higher frequencies the balance is between non-linear
interactions and Marangoni dissipation, which is only slightly depending on fetch. Note that, as clearly represented
in Figure 12, the attenuation is maximum around 3 Hz (smaller than the resonance frequency) and the maximum
damping ($D$) of wind wave energy (see for comparison Figure 6) is two or three orders of magnitude smaller than
with only wind waves (experiments W06). This is the consequence of two parallel facts: less wind wave energy in
presence of paddle waves, and a decreased efficiency of damping by oil film, as it is evident comparing the right

panels of Figure 2 and Figure 9. These effects have impact also on the short wind waves, as the damping effect

appears to cease at frequencies higher than 9 Hz (Figure 12)

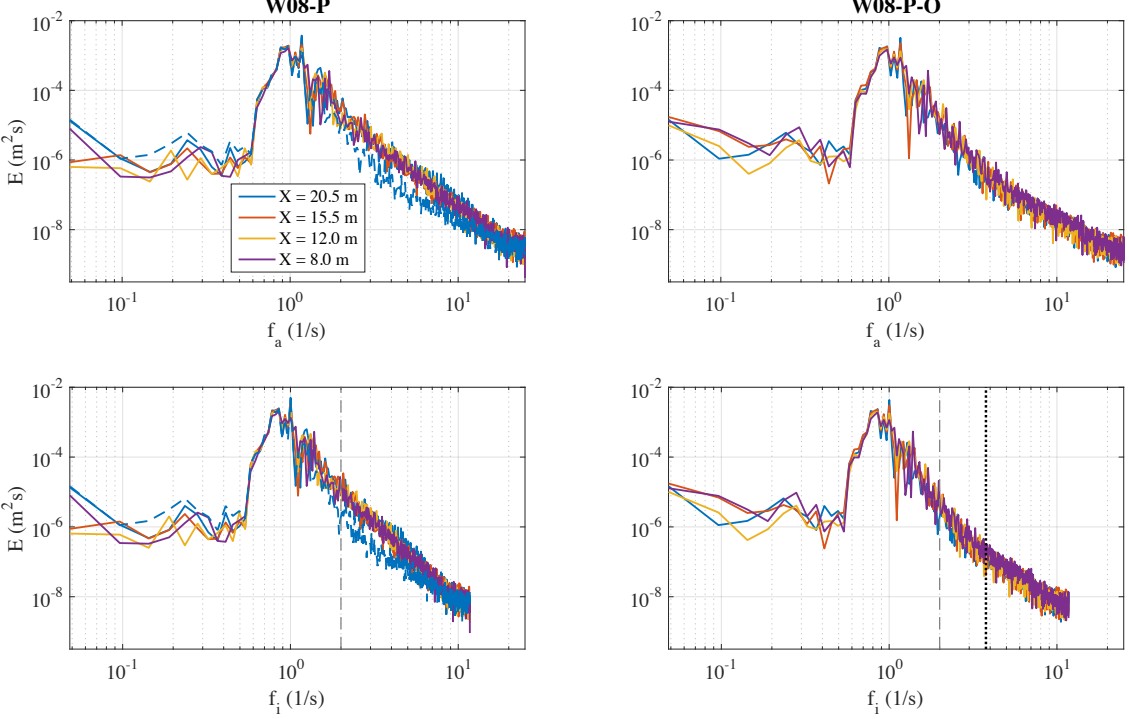

**Figure 11: Wave energy spectra $E(f_a)$ and $E(f_i)$ of the water surface elevation $z(t)$ at different fetches for the reference wind speed of 8 m/s and irregular paddle waves. (left) Clean water experiment W08-P. (right) Water with oil experiment W08-P-O (the spectrum at the fetch $X = 20.5$ m is replicated with a dashed blue line on the left panel). On bottom panels the dashed grey vertical line shows the maximum frequency (2 Hz) produced by the paddle, and the dotted black vertical line on the bottom-right panel shows the fish oil resonance frequency $f_{res} = \omega_{res}/(2\pi) = 3.77$ Hz.**

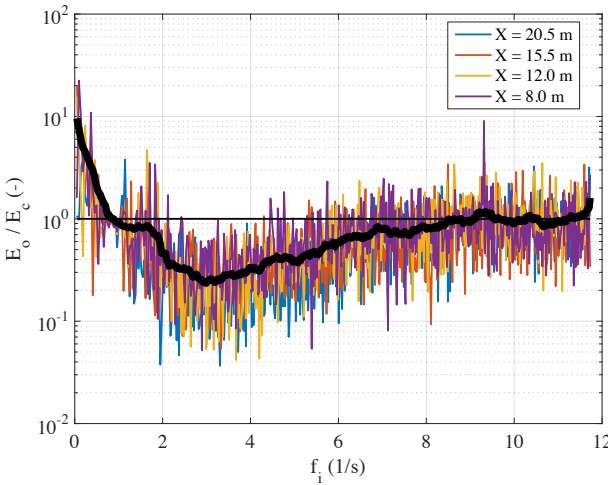

**Figure 12: Damping of the wave energy for oil-covered water surface. The damping coefficient is evaluated as the ratio $D = E_o / E_c$ between the variance density spectrum of the water surface elevation with oil slick ($E_o$; experiment W08-P-O) and in clean water ($E_c$; experiment W08-P). The solid black thick line is the average shape of the damping coefficient (for the sake of clarity the series is smoothed with a moving average procedure). The solid black thin line shows the level $E_o = E_c$.**

Similarly to what was done for the only wind-wave case (Figure 7), we check the small difference of energy with fetch of the two basic components (paddle and wind waves) of the spectra in clean water and in water with oil. The differences are small and are visible in Figure 13. Because of the partial superposition of the two (paddle and wind) frequency ranges, we have computed, for the intrinsic spectra, the surface elevation variance below 1.3 Hz (PW in Figure 13, dominated by paddle waves) and above 2 Hz (WW in Figure 13, dominated by wind waves). For the high-frequency part of the wind wave spectrum, in clean water, waves grow slightly with fetch, gaining 30% energy passing from $X = 8$ m fetch to $X = 20.5$ m. The growth is obviously much smaller than in absence of swell. The presence of oil (red marker) makes the waves progressively decreasing with fetch (the coefficient $\beta < 0$), consistently with, and with the same explanation for, the results obtained without paddle waves.

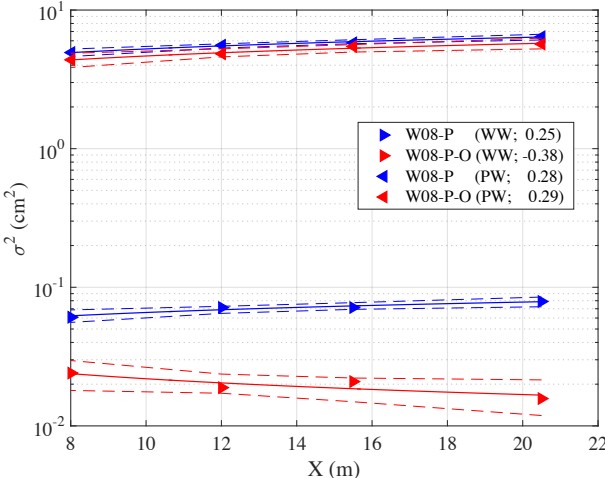

Figure 13: Spatial evolution of the water surface elevation variance $\sigma^2$ of co-existing wind (WW) and paddle (PW) waves in clean

water (W08-P) and with oil slick (W08-P-O). The solid lines show the power law-type functions fitting the experimental data. In the

legend the coefficient $\beta$ of the fitted power law is shown. The dashed lines show the prediction bounds (confidence level of 90%) for

the fitted curves.

The growth of paddle waves under the wind forcing appears to be marginally affected by the presence of

oil (PW markers in Figure 13), generalizing to a more realistic wave field the results of Mitsuyasu and Honda

(1986) obtained for monochromatic paddle waves. We interpret this arguing that the reduction of surface roughness

between W08-P and W08-P-O is not sufficient to change substantially the vertical profile of the turbulent airflow,

hence the generation process acting on long waves. This is confirmed by Figure 14 showing the wind profile with

and without oil. There is only a small difference between the two cases, however the friction velocity is, as

expected, larger in clean water. This implies that the minor disturbances we see in the right panel of Figure 9

suffice for making the wind feel the surface as rough. Indeed we are here at the limit because the further, almost

complete, wave reduction we see in Figure 2 for the experiment W06-O changes dramatically the wind profile, as

already seen in Figure 3.

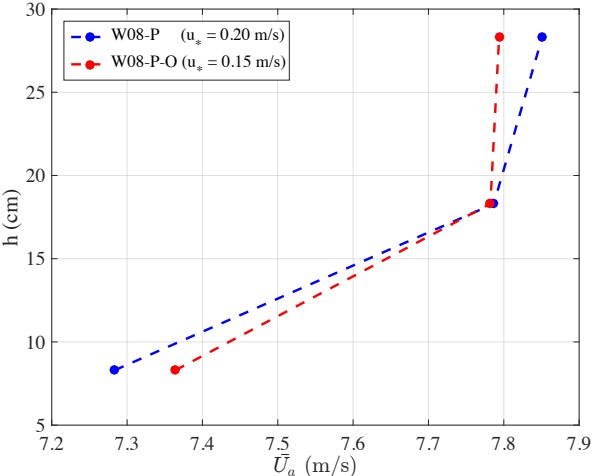

Figure 14: Vertical profile of the wind velocity component $\overline{U}_a$ measured over the water surface at fetch $X = 11.5$ m for co-existing wind- and paddle-generated waves in clean water condition (blue) and for oil-covered surface (red). In the legend, the value of the friction velocity is shown within brackets. Only the value recorded at the three lowest Pitot tubes are shown.

## 4  Discussion

Our previous description of the general methodology and results has been focused much on the implications of having the water surface covered by a very thin (~$10^{-8}$ m) layer of fish oil. However interesting in itself, it is clear that the main virtue of these experiments has been the opening of new perspectives on the physics of air-sea interactions. We discuss here the main suggestions and ideas derived from our results.

– The different wind profile and wave growth without and with oil clearly show that the stress felt by the atmosphere is, as anticipated by Janssen (1991), the sum of the friction stress and the input to waves. Lacking the latter, the atmospheric stress reduces to the purely frictional one. This has implications also for circulation modeling where quite often the wave intermediate role (wind input to waves followed by wave input to current via breaking) is by-passed by an artificially inflated surface friction to current. In this respect, McWilliams and Restrepo (1999) showed how the general global circulation could be obtained also driving it only with breaking wave momentum injection.

– Especially with young, hence relatively slow, waves it is essential to consider generation with respect to the surface current. In our case, lacking any instrument to measure current, and in particular its vertical profile, we have used two independent methods based on video documentation to get an estimate of the surface current drift. The two approaches provided similar results, consistent with the one derived from the wind speed and

similar conditions in the literature. Lacking any data on the current vertical profile, and supported also by the recirculation characteristics of the wave tank, we made the blunt assumption of a vertically uniform current. This is a first order approximation, that we consider acceptable in our case because the related discussion concerns only very short waves. In any case, more complete experiments are planned for the near future.

– We have found it interesting to look at how the water surface reacts and evolves under the action of an impulsive wind forcing. For this first analysis of the obtained data, we have limited ourselves to the steady, fetch-limited conditions. This subject takes us to a very short discussion on the generation mechanism(s) of the earliest waves. In this respect, the spectral approach to wind wave modeling began with the study by Pierson et al. (1955), followed by the two parallel and independent, but complementary, papers by Phillips (1957) and Miles (1957), and the definition of the energy balance equation by Gelci et al. (1957). While the Miles' mechanism, refined by Janssen (1991), provides the bulk of the input to waves, we still need to trigger the first wavelets on which Miles, and non-linear interactions, can then act. Two processes compete for this first stage: the just mentioned one by Phillips, associated to assumed pressure oscillations moving with the wind, supported also by the recent paper by Zavadsky and Shemer (2017), and a sort of Kelvin-Helmholtz instability (Kawai, 1979) due to the strong vertical shear in the surface water layer following the initial action by wind. The matter is not relevant for practical purposes because these initial stages are usually parameterized or bypassed in wave modeling in a pragmatic way. We know these wavelets appear and their exact dimensions are irrelevant for the following evolution of the actual field. In this regard, our experiments provide a small piece of information. The Phillips' mechanism is supposed to act on any wavelength, independently of the other ones. We argue that in the oily experiment, granted the dissipation at the wavelet scale due to the Marangoni effect, nothing would impede the Phillips' mechanism to act on the longer waves. However, we found no evidence of energy in the corresponding wave components.

– With short wind and long paddle waves, the presence of oil still reduces wind-wave generation, but there is more wind-wave energy than with only wind. Therefore the presence of a swell reduces the effectiveness of the oil in impeding local generation. We hypothesize this as mainly due to the long wave orbital motion that detunes the resonance conditions between Marangoni waves and gravity-capillary waves. Moreover, this motion continuously disrupts the continuity of the oil layer and hence the effectiveness of the Marangoni forces.

# 5 Conclusions and summary

With the help of an experimental facility, we have studied the influence of a very thin layer of fish oil on wind and paddle waves as well as on the parameters of the lowest airflow layer. Measurements of sea surface elevation at different fetches and of wind speed were carried out in both clean water and in water with fish oil producing a viscoelastic film on the surface. The damping of short gravity-capillary waves by surfactants appears a convenient condition to study, to a large extent, the processes of interaction between the water body and the atmosphere. The aim of the present study is thus to evaluate the influence of the fish oil film on the growth of wind and paddle waves and on the air-water interaction process. Taking this viewpoint, the principal conclusions of our study can be summarized as follows:

– Marangoni forces, associated to the presence of the fish oil, quickly dissipate and impede the formation of the first wavelets, hence, in a laboratory, the growth of any wind sea. As it is generally agreed, this dissipation at short wavelengths leads to an intensified energy transfer via non-linear interactions from the bulk of the spectrum, in so doing not only smoothing, but also partly calming the sea. In the fish-oil covered wave tank, the powerful suppression of the first wavelets leaves the airflow vertical profile unaffected by the wave field, so that the Miles-Janssen wave growth mechanism is not triggered.

– Our results show the efficacy of the fish oil in suppressing the wave generation by wind. Indeed, in the experiments in a wind-wave tank contaminated by non-animal surfactants, Hühnerfuss et al. (1981) found that the peak of the spectra is shifted towards higher frequency when compared with clean water conditions. However, a wind-wave spectrum was still present in their tests, whilst we have found that, using fish oil in the wind-only condition (reference wind speed of 6 m/s), the wave field does not grow from the rest condition, leading to a strong modification of the airflow vertical profile.

– The experiment with wind-wave only in water with fish oil (experiment W06-O) gave us the unique opportunity to investigate (albeit preliminarily) the interactions at the air-water interface in absence of surface waves. Clearly, the stress exerted in that case by the airflow is smaller than the stress in clean water (when the wave field regularly develops) and the water current is determined by the wind shear only, so that $\tau_w \approx \tau_a$. This condition is expected to be different from the condition obtained for fully developed waves (in clean water), when one should find that in Eq. ( 7 ) the momentum flux into the water column $\tau_w$ should become the

atmospheric stress $\tau_a$ as the wave field reaches equilibrium (see ECMWF, 2017). For instance, the water current vertical profile below the surface is expected to be different in clean water and in water with oil. In this respect, new experiments are already planned that will investigate also the drift current distribution beneath the water surface for an oil-covered water surface.

- The growth of paddle-generated long waves seems to be little affected by the roughness of short waves. Provided a minimal background of very short waves (in practice surface roughness) is present, the growth of paddle waves under the action of wind is largely independent on the background roughness, a fact we ascribe to the similar distortion of the wind profile in the two cases.

With our study we have shown that wave generation and dissipation, and more general atmosphere and sea dynamic interaction, are still to be fully explored. The approach we followed, experiments in a wind wave tank without and with oil on the surface, offers new possibilities for studying this old, but still fruitful, field.

## Appendix A: The Marangoni wave damping effect

The physics of the Marangoni effect is well understood, and it is generally agreed that the gravity-capillary wave energy is damped by a surfactant by the following mechanism. Immediately after a wave field enters a surfactant patch, the Marangoni effect due to the viscoelastic film on the water surface increases viscous dissipation of short wave energy at specific frequencies. This is caused by an additional dissipation in the surface boundary layer. Indeed a shear stress in the boundary layer is required to balance the stress due to the surface tension gradients originated by the non uniform concentration of the surface viscoelastic film (alternatively contracted and expanded due to the passage of a wave). This gives rise to additional longitudinal waves in the boundary layer superimposed on the existing surface waves that are thus damped by a resonance-type mechanism. The dissipative spectral sink (the so-called Marangoni resonance region) is at frequencies between 3 Hz and 8 Hz, the scale of the maximum damping depending on the dilatational modulus of the surfactant (Alpers and Hühnerfuss, 1989; Fiscella et al., 1985). The resonant angular frequency $\omega_{\text{res}}$ for the Marangoni-force wave damping is given by

$$\omega_{\text{res}} = \left\{ \frac{[\cos(\pi/8)]^4 \, g^4 (\eta \rho_w)}{\varepsilon^2} \right\}^{\frac{1}{5}} (\text{rad/s}) \quad (10)$$

where $g$ (m/s$^2$) is the gravity acceleration, $\rho_w$ (kg/m$^3$) the clean water density, $\eta$ (Ns/m$^2$) the clean water dynamic viscosity, and $\varepsilon$ (N/m) denotes the dilatational modulus of the surface film. As stated by Eq. ( 10 ), the larger the

modulus $\varepsilon$ the longer the damped waves. The maximum damping of gravity-capillary waves is attained at a
frequency lower than $\omega_{\text{res}}$ (the Marangoni wave is a strongly damped wave) and the width of the resonant damping
is quite broad (the half-power width is on the order of 1 to 2 Hz; Alpers and Hühnerfuss, 1989).

4       In a more formal approach, the description of water surface gravity waves follows generally a statistical

approach by means of the development of the wave elevation variance spectrum $E = E(k, \theta; x, t)$ in the physical
space $x$ and time $t$ ($k$ is the wavenumber and $\theta$ the propagation direction), its evolution governed by the energy
balance equation (Gelci et al., 1957). In deep waters, it reads

$$\frac{\partial N}{\partial t} + \frac{\partial N}{\partial x} \cdot (\dot{x}N) = S_{\text{in}} + S_{\text{nl}} + S_{\text{di}} \qquad (\,11\,)$$

where $N = E / \omega$ is the wave action density spectrum with $\omega$ the intrinsic angular frequency. Furthermore,
$\dot{x} = c_{\text{g}} + U$ with $c_{\text{g}}$ the wave group velocity and $U$ an appropriate current. The right hand side of Eq. (\,11\,)
represents the net effect of sources and sinks for the spectrum (Komen et al., 1994): $S_{\text{in}}$ is the rate of energy
transferred from the wind to the wave field, $S_{\text{nl}}$ is the rate of nonlinear energy transfer among wave components
with different wavenumber, and $S_{\text{di}} = S_{\text{di,b}} + S_{\text{di,v}}$ is the rate of energy dissipation due to breaking ($S_{\text{di,b}}$) and viscous
forces ($S_{\text{di,v}}$).

14       In our experiments the observations dealt with in this paper are the ones collected at steady state, so that

the spectrum at any fetch is determined by the upwind evolution of the source functions $S_{\text{in}}$, $S_{\text{nl}}$ and $S_{\text{di}}$, that is

$$N = \int_0^x \left[ \left(c_{\text{g}} + U\right)^{-1} (S_{\text{in}} + S_{\text{nl}} + S_{\text{di}}) \right] dx \qquad (\,12\,)$$

where, with good approximation, we have neglected the cross-tank wave energy evolution. The velocities $c_{\text{g}}$ and $U$
are also, but very weakly, fetch dependent. The balance in Eq. (\,12\,) indicates that any modification of the wave
energy may and must be caused by changes in the rate of wind input, dissipation, or/and nonlinear transfer.

19       In presence of oil, all three source functions undergo a change compared to the clean water condition. Indeed,

the rapid suppression of short waves by Marangoni forces reduces the water surface mean slope, which leads to a
change of the wind vertical profile and rapid decrease of the momentum flux from the wind to the wave field (see,
e.g., Mitsuyasu and Honda, 1986). Those two effects combined produce a change of the shape of the wave
spectrum in the equilibrium range, which leads, via nonlinear wave-wave interaction (Hasselmann, 1962), to a slow
leakage (but fast compared to the pure viscous one; Alpers and Hühnerfuss, 1989) at wavelengths longer than those
at which the Marangoni forces are effective
Given the possible variability of the surfactant density on the water surface, it is natural to wonder about the
related sensitivity of the effect on waves. Analysing the wind-wave tank experiments with surfactants (sodium
lauryl sulfate) presented by Mitsuyasu and Honda (1986), we can distinguish two different regimes for wave
attenuation, which correspond to weak and strong wave damping, respectively. Firstly, for small surfactant
concentrations (i.e. weak damping), the peak frequency of the wind-wave spectrum in presence of films is shifted
to higher frequencies relative to the peak frequency of clean water (in other words waves develop more slowly).
However, spectra preserve $u_*$-similarity, and the new spectral shape was ascribed mainly to the decrease of the
wind stress: the surfactants smooth the surface and act to reduce the wind stress, therefore the waves grow less. We
interpret this result assuming that for low surfactant concentrations the effectiveness of the Marangoni damping is
small (i.e. the water surface is not fully covered by an uniform film), but with a partially reduced aerodynamic
roughness. On the contrary, for the highest concentration (hence with a strong damping) the similarity no longer
holds, and most of the energy around the peak is lost (the maximum energy is at a frequency smaller than the one in
clean water; see also Figure 5 and Figure 6).
A question about Marangoni forces concerns the maximum wind speed for which they are expected to keep
their damping efficiency. In early studies, disappearance of the Marangoni damping was observed by Mitsuyasu
and Honda (1986) for wind speed larger than a critical value, that those authors found to be 12.5 m/s. A possible
explanation is provided in the study by Alpers and Hühnerfuss (1989), who argued that above a certain friction
velocity (around 0.5 m/s) the Marangoni dip is filled in owing to a large flux of wave energy into the Marangoni
resonance region by nonlinear wave-wave interactions. In addition, for stronger wind stresses the viscoelastic film
becomes more mixed with the underlying clean water bulk (the film is "washed down"), so that the Marangoni
damping is strongly attenuated (see also Feindt, 1985). However, these conclusions seem not to be fully consistent
with the recent analysis made by Cox et al. (2017) of the saving in 1883 of the crew of a sinking vessel by the ship
*Martha Cobb* under very severe stormy conditions (wind speed around 20 m/s). In that occasion 19 litres of fish oil
were dribbled into the sea and the log records report what, after a 20-minute delay, was defined as a "magic effect",
i.e. that the water surface smoothed and breakers disappeared around the vessel, allowing the crew to be saved
using a small open deck dingy. Cox et al. (2017) estimated that after 20 minutes the surface covered by the oil film
was about 0.4 km$^2$, hence, the average oil thickness was about $5\times10^{-8}$ m. In those conditions, therefore, the
thickness was comparable to that used in our experiments in the tank and, in spite of the high wind speed, the wave
damping, with practical cancelling of wave breaking, was still effective.

## Appendix B: Surface current drift estimate using optical flow

As it is specified in Section 3.1.1, tiny bubbles moving on the surface along the tank, and visible in the 1920x1080
pixel images captured at 60 Hz by a video-camera placed outside the tank close to prove G4, made it possible to
have an estimate of the water surface current. The probe's two vertical wires, whose measures have been accurately
determined, were used to map visual features from the image space (in pixels) to the tank surface space (in metres).
To ease the computation, we manually defined a quadrilateral area in the image and computed the homographic
transformation between the quadrilateral space to a normalized rectangular space of size 512x512 pixels. To
account for the possible uneven illumination along the sequence, each image was normalized so that the intensity
values have zero mean and unitary standard deviation. Due to the optical characteristics of the water, bubbles are
not the only visual features present in the frames. Indeed, visual clutter mostly due to reflections makes it more
complex to reliably track the bubbles during the whole sequence. Since the cameras were firmly placed on a tripod
during the acquisition, and light conditions were mostly controlled, the clutter appearance remains quite stable
among the frames, with slight fluctuations due to the small waves and the automatic exposure adjustments of the
camera.
Therefore, we performed a simple background subtraction by computing the squared difference between
each frame and the frame obtained by averaging the intensity values of the previous 3 frames. To remove the high-
frequency noise and artefacts caused by video compression, we blurred the background-subtracted image with a
3x3 Gaussian kernel and applied a threshold of 1.8 to obtain binary images. From the binary image of each frame,
we extracted the location of each particle by using the function *goodFeaturesToTrack()* provided by the *OpenCV*
Computer Vision Library (Bradski and Kaehler, 2008). We specified 2000 max corners, a quality level of 0.08, a
minimum distance between features of 3 pixels and a block size of 9x9 pixels. Then, we computed the sparse
optical flow with respect to the subsequent frame at the location of each particle using the iterative Lucas-Kanade
method with pyramids provided by *OpenCV*. We used a 15x15 pixels window for the matching and a pyramid
depth of 5 levels. The computed optical flow gives the amount of movement performed by each tracked particle
between each frame (Figure 15). By knowing the mapping between pixels and tank metric space, and the camera

frame rate, we could estimate the speed (in m/s) of each particle. So, we transformed particle locations and movement vectors back to the original image space by inverting the homography.

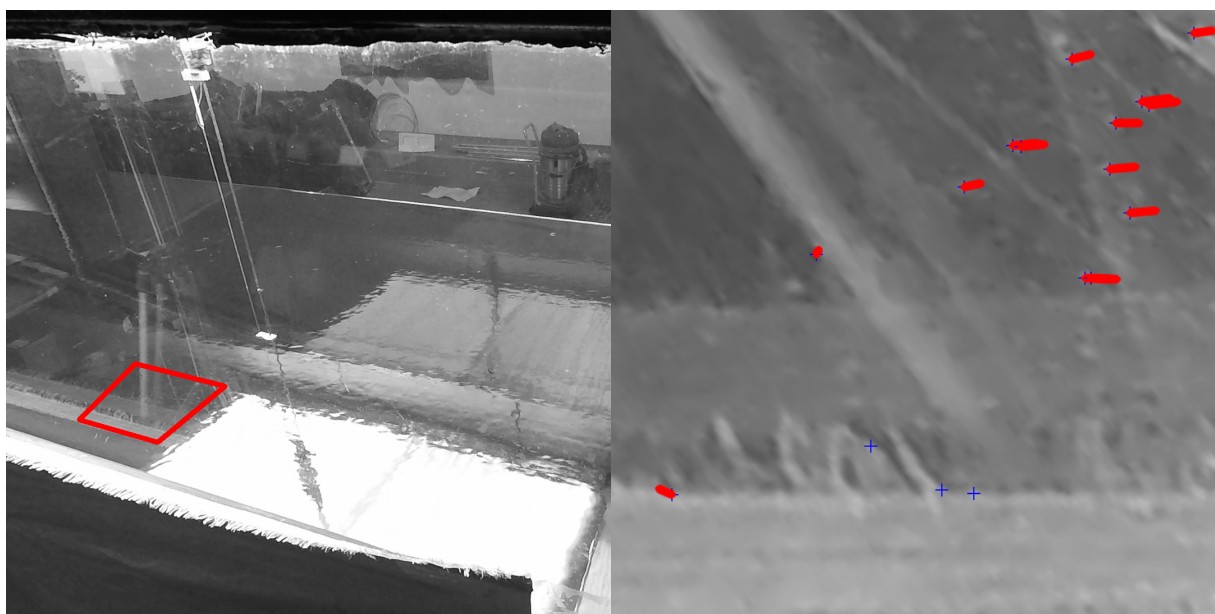

**Figure 15: Estimate of surface current speed by optical flow during experiment W06-O. (left) Bounded by a red polygon, the surface area used for the determination of the flow. (right) Example of detected particles with their corresponding movement (wind blowing from left to right of the pictures).**

## Supplementary Material (Videos)

1. Supplementary material SM1 available at https://doi.org/10.5281/zenodo.1434262.

2. Supplementary material SM2 available at https://doi.org/10.5281/zenodo.1434272.

**Author contributions**. All authors have contributed to the writing of the paper. Experiments were planned by LC, FQ, and AB, and executed by all authors. Data analysis and interpretation were performed by AB with the help of LC, FB, and SJ.

**Competing interests**. The authors declare that they have no conflict of interest.

## Acknowledgements

Authors gratefully acknowledge the funding from the Flagship Project RITMARE - The Italian Research for the Sea - coordinated by the Italian National Research Council and funded by the Italian Ministry of Education, University and Research within the National Research Program 2011-2015. Luigi Cavaleri has been partially

supported by the EU contract 730030 call H2020-EO-20116 'CEASELESS'. Fanqli Qiao was supported by the international cooperation project on the China-Australia Research Center for Maritime Engineering of Ministry of Science and Technology, China under grant 2016YFE0101400, and the international cooperation project of Indo-Pacific ocean environment variation and air-sea interaction under Grant GASI-IPOVAI-05. The authors are pleased to acknowledge useful discussions with P.A.E.M. Janssen, Norden Huang, Jean-Raymond Bidlot, and Francesco Barbariol. We are grateful for the efforts of the editor and reviewers whose insights greatly improved this paper.

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
