# Peer review of "Analysis of the effect of fish oil on wind waves and implications for air-water interaction studies"

_Ocean Science, 2018_

## Referee Comment (RC1) · Anonymous Referee #1 · 13 Dec 2018

The manuscript "Fish oil in a wave tank: a look at the air-water response" is not suitable for publication in the present form because the text lacks clarity and structure to such a degree that I find it impossible to evaluate the quality of the underlying scientific work. It is also not clear how the work relates to previous studies of the past 30 years, and what any potential new findings are.

The first part of the introduction (line 1-30) do not communicate relevant information about the scientific background or this study but instead seem more like a philosophical discussion about some scientific methods, for instance, line in 18-23:

"The interest is not only on the results of the experiments, but on the physics they reveal and the considerations they allow on the general problem of wind wave generation. Following this logical link we have carried out a series of experiments aimed at, if not

solving the whole problem (a daunting task), at least shading new light on some of its aspects. Science proceeds often by negations. New results may not only hint in one direction, but also exclude a solution, in so doing helping focusing along the right path."

Section 1.1 then proceeds with text on the physical background, but does not prepare the reader for the work that follows. For instance, large parts of the paper seem to be about Marangoni damping. In line 17 in section 1.1, the author write "This resonance-type Marangoni damping (to be soon described)..." , but it is never described in the following text. The next time Marangoni damping is mentioned, the authors already assume that the reader is familiar with it.

The ongoing discussion in the introduction jumps form topic to topic without a continuous thread the reader could follow. The authors do not present what the motivation for this study is (is the goal to "still angry waters"?), nor do they present a knowledge gap that they want to address or a goal for this study.

Section 3 (the results) continues to provide introductory material, instead of focusing on presenting the results. I doubt that all this information is needed to discuss their experiments and results. Background knowledge, results, and discussions are intertwined throughout the entire result section which makes it very difficult to read.

The discussion section does not establish a link to previous works, e.g. experimental studies on oil in a wave tank or similar. The authors claim that they provide a now look at air-sea interaction, but it is not clear to me what the new findings are. In the conclusion section, one finally learns what the aim of the study is ( to evaluate the influence of the oil film on the frequency spectrum and growth of waves), but I doubt that these are new finding and they should be discussed in context with recent studies.

Beside the quality of how the work is presented, I am also in doubt if new knowledge is provided in this study. Most of the presented references on water-oil-wave interaction are more than 30 years old, and the conclusion that an oil film dampens wind-waves but not swell is considered general knowledge. It should be made clear what new findings

is, and what confirmations of previous experimental or theoretical results are.

It seems to me that the experiments carried out for this study are of good quality, and that they can provide useful results. However, the paper should be entirely re-written, focusing on the experiments and results instead of philosophical questions and needles history details.

———————————————————

---

## Referee Comment (RC2) · Anonymous Referee #2 · 18 Dec 2018

In this article the authors studies the impact of fish oil surfactant on the generation of surface gravity wave and on its feedback on the wind profile. To this purpose they use an experimental setup. They shows that when surface wave are inhibited by fish oil, the wind induce surface current is very similar to the one with surface waves. The overall document read wells and the scientific methods and experimental set up seem robust, nevertheless is it difficult from the introduction to understand the real motivation and scientific questioning of the paper which can be seen as a enumeration of different statements and results which would need to be articulated and discussed in regard to a clear scientific questioning . Therefore, although i recognize the quality of the work and the potential of the papers results i recommend to better focus the aim of the works and there implication for ocean atmosphere exchanges before publication of the

manuscript.

---

## Author Comment (AC1) · 15 Feb 2019

**Revision of the manuscript: os-2018-111**

**Fish oil in a wave tank: a look at the air-water response**

Alvise Benetazzo, Luigi Cavaleri, Hongyu Ma, Shumin Jiang,
Filippo Bergamasco, Wenzheng Jiang, Sheng Chen, and Fangli Qiao

Please find enclosed a detailed list of replies to the Referee 1 comments on the paper in Subject.

We would like to thank the Referee for the comments and suggestions, which have been useful to improve the manuscript.

Sincerely yours,

**Alvise Benetazzo**
Institute of Marine Sciences
National Research Council
Venice, ITALY

**REFEREE #1**

We would like to thank Anonymous Referee #1 for the constructive and helpful comments. Our responses to the comments (in *Italic*) are given below.

AUTHORS: We thank the reviewer for pointing out that the experiments carried out for this study are of good quality and that they can provide useful results. However, to account for the criticisms raised by the reviewer, the manuscript has been partially reshaped and rewritten. Below, our replies to the specific comments are provided.

REFEREE: *The manuscript "Fish oil in a wave tank: a look at the air-water response" is not suitable for publication in the present form because the text lacks clarity and structure to such a degree that I find it impossible to evaluate the quality of the underlying scientific work. It is also not clear how the work relates to previous studies of the past 30 years, and what any potential new findings are. The first part of the introduction (line 1-30) do not communicate relevant information about the scientific background or this study but instead seem more like a philosophical discussion about some scientific methods, for instance, line in 18-23: "The interest is not only on the results of the experiments, but on the physics they reveal and the considerations they allow on the general problem of wind wave generation. Following this logical link we have carried out a series of experiments aimed at, if not solving the whole problem (a daunting task), at least shading new light on some of its aspects. Science proceeds often by negations. New results may not only hint in one direction, but also exclude a solution, in so doing helping focusing along the right path." Section 1.1 then proceeds with text on the physical background, but does not prepare the reader for the work that follows. For instance, large parts of the paper seem to be about Marangoni damping. In line 17 in section 1.1, the author write "This resonance- type Marangoni damping (to be soon described)..." , but it is never described in the following text. The next time Marangoni damping is mentioned, the authors already assume that the reader is familiar with it. The ongoing discussion in the introduction jumps form topic to topic without a continuous thread the reader could follow. The authors do not present what the motivation for this study is (is the goal to "still angry waters"?), nor do they present a knowledge gap that they want to address or a goal for this study.*

AUTHORS: We thank the reviewer for this keen comment that helped to improve our paper. Indeed, we agree that in the original manuscript the description of the purpose and originality of our research could be improved.

Following the reviewer's suggestions, in the revised paper we have hence highlighted in the "Introduction" the motivations of our study. Firstly, we aim at addressing the general question on how a surfactant with high dilatational modulus significantly alters the generation of wind waves and consequently the momentum transfer across the air-water interface. Secondly, we verify for the first time the effectiveness of polar fish oil, often empirically used in the past to smooth angry seas, in suppressing wind waves. Finally, we show the benefits and potentiality of comprehensive laboratory studies with surfactants to explore the interactions between air and water. Among the original results we present, for the first time we document that with an effective surfactant the wind-wave field does not grow from the rest condition, but a surface water drift is still present determined by the wind shear only. To make the motivations clearer, as suggested, we have hence reshaped abstract, introduction (most of the section 1.1 has been moved to Appendix A "The Marangoni wave damping effect"), text (e.g. the description of the interaction between short and long waves has been moved to the Introduction), and modified the title that now reads as "Analysis of the effect of fish oil on wind waves and implications for air-water interaction studies".

REFEREE: *Section 3 (the results) continues to provide introductory material, instead of focusing on presenting the results. I doubt that all this information is needed to discuss their experiments and results. Background knowledge, results, and discussions are intertwined throughout the entire result section which makes it very difficult to read.*

AUTHORS: In Section 3, material pertaining to the results and their interpretation is presented. Having clarified in the Introduction the purposes of the present study, this section is better framed in the flow of the manuscript. Some modifications have been however made to Section 3, moving, as suggested, part of the material to Section 2 describing the experiments.

REFEREE: *The discussion section does not establish a link to previous works, e.g. experimental studies on oil in a wave tank or similar. The authors claim that they provide a now look at air-sea interaction, but it is not clear to me what the new findings are. In the conclusion section, one finally learns what the aim of the study is (to evaluate the influence of the oil film on the frequency spectrum and growth of waves), but I doubt that these are new finding and they should be discussed in context with recent studies. Beside the quality of how the work is presented, I am also in doubt if new knowledge is provided in this study. Most of the presented references on water-oil-wave interaction are more than 30 years old, and the conclusion that an oil film dampens wind-waves but not swell is considered*

*general knowledge. It should be made clear what new findings is, and what confirmations of previous experimental or theoretical results are.*

AUTHORS: To the best of our knowledge, the oil effect in smoothing angry seas is not fully understood. While the Marangoni damping is well described and its modification on the wave spectrum can be predicted (see e.g. Alpers and Hühnerfuss, 1989), the whole interplay between wind shear stress and surface wave source functions is still and open question (Cox et al., 2017. "Oil has been known since antiquity to still angry seas, but how it does so remains mysterious" by Cox, C. S., Zhang, X. and Duda, T. F., 2017. Suppressing breakers with polar oil films: Using an epic sea rescue to model wave energy budgets, Geophys. Res. Lett., 44, 1414–1421). This fact largely stimulated our research. Moreover, the references we mention trace back to 30 years, because since then no thorough investigations have been made on the suppression of wind waves using surfactants (neither in open sea nor in a laboratory). Our study was thus aimed at verifying for the first time the fish oil effect on the generation of wind waves in a tank, but also at exploring the potentiality of this peculiar experiments to investigate the air-sea interaction processes. We mention the original analysis we made on the surface current that was observed and measured in absence of waves (almost fully damped by the oil slick) and the investigation of the interplay between long and short waves in receiving wind input in clean water and with suppressed short waves by oil.

REFEREE: *It seems to me that the experiments carried out for this study are of good quality, and that they can provide useful results. However, the paper should be entirely re-written, focusing on the experiments and results instead of philosophical questions and needles history details.*

AUTHORS: We thank the reviewer for pointing out the overall merit of our study.

[revised manuscript text omitted]

Wind blowing over the water surface generates wind waves and drift currents. It is instructive that the physics of this evident truth is still a question of debate. One reason is that the implied physics spans a large range of scales, the various processes interacting among them, and possibly hiding the reasons of that behavior. Nature offers a full panorama of events at all the possible scales. However, it is mainly in the laboratory that we can explore, also with the desired repetitiveness, the details of some, albeit limited in scale, processes. Indeed this approach has provided in time enlightening findings to be then used in the daily operational activities. Already in the '70s and '80s Mitsuyasu, in a series of remarkable papers, provided basic hints into the generative and dissipative processes of wind waves (see, among others, Kusaba and Mitsuyasu, 1984; Mitsuyasu, 1966; Mitsuyasu and Honda, 1982, 1986). Mark Donelan, using data first from a tower in lake Ontario and then in a laboratory wind wave tunnel in Miami, provided basic hints in several aspects of wind wave generation (see e.g. Donelan, 1990). In more recent times, following a very sophisticated and detailed series of experiments, Buckley and Veron (2016) have provided a detailed description of the air flow during wave generation. The specific problem of the trigger of the initial wavelets has been dealt with by Kawai (1979), van Gastel et al. (1985), and more recently by Liberzon and Shemer (2011) and Zavadsky and Shemer (2017).

A problem that (in most of the cases) does not concern the open ocean is how air and sea interact when the water surface is covered by a thin layer of surface active agent. This physical aspect has been early dealt with experimentally in studies by Hühnerfuss et al. (1981) and Mitsuyasu and Honda (1986). The interest is not only on the results of the experiments, but on the physics they reveal and the considerations they allow on the general problem of wind wave generation. Following this logical link we have carried out a series of experiments aimed at, if not solving the whole problem (a daunting task), at least shading new light on some of its aspects. Science proceeds often by negations. New results may not only hint in one direction, but also exclude a solution, in so doing helping focusing along the right path.

In the following we describe what has been done, for each experiment stressing the doubts and the implications. Given that a large part of what was done deals with fish oil on the surface, following subsection 1.1 provides a compact, but sufficient for the purpose, description of the related physics. The general description of the experimental set-up is in section 2, where we also list the general plan of the experiments and the finally available data. The actual technical description of the results of the experiments are in section 3, the whole then discussed in section 4, and summarized and itemized in the final section 5.

**1.1 A little physics on the interaction between oil slick and gravity-capillary waves**

It is well known that the addition of a thin film (from $10^{-9}$ to $10^{-8}$ m, almost mono-molecular) of surfactant (blend of *surface active agent*) to the water surface has a intense effect on the energy of the gravity-capillary waves by altering the surface tension at the water-air interface (Fiscella et al., 1985). Oil has been used for centuries to smooth the sea surface, so much that expressions as "to pour oil on troubled water" have acquired a more general meaning. See later in the paper the impressive example reported by Cox et al. (2017). Crucial in this respect is the type of oil, in particular its polarity. Mineral oils, so often wrongly used during the Second World War, are not effective because their molecules tend to group together in a heap. On the contrary the polar molecules of fish, and partly also vegetable, oils repel each other. Hence, once poured on water, they tend to distribute rapidly on the available surface acquiring a quasi-monomolecular level. Known since ancient time, this effect was first studied in the 19th century by the Italian physicist Carlo Marangoni, hence the official name of the process (Marangoni, 1872). In relatively recent times the first report of Marangoni damping of wave spectra came from Cini et al. (1983) who had noted the effect in the polluted (although by mineral oils) water in the gulf of Genoa, Italy. However, clear evidence of Marangoni damping on slick-covered ocean waves was first presented by Ermakov et al. (1985, 1986) during field experiment in the Black Sea.

This resonance type Marangoni damping (to be soon described) can be effective for surface waves in two possible conditions. The first one is in the open ocean, in which an existing wind-forced wave field travels trough a surfactant patch, with the consequent possibility of detection (lack of return signal) by microwave radars (Feindt, 1985). The second one, typical of laboratory experiments, is where the wind blows over a water surface homogeneously covered (since the rest condition) with a surfactant film. The latter is the one we dealt with in the experiments described in this study.

The presence of an extremely thin (practically mono-molecular) layer of oil on the surface strongly affects the air-sea interaction. In this respect, it is generally agreed that, in clean water, the growth of the first detectable ripples on the water surface is rather well explained by the effect of air turbulence advected by the wind (Phillips, 1957). That process is quickly overtaken as the waves grow by the feedback caused by the wave-induced pressure oscillations in the air, as soon as the airflow vertical profile is modified by waves. Miles (1957) proposed a wave growth mechanism that accounts for this change. This theory was extended and later applied by Janssen (1991) to wave forecasting. Its validity is questioned for very short waves whose phase speed is as slow as the air friction velocity (Miles, 1993). According to the shear-flow model by Miles, waves with phase speed $c$ grow when the curvature in the vertical wind profile, at the height (called critical height) where the wind speeds equals $c$, is negative. As a result, the wind profile changes because of the continuous transfer of energy to the waves (Janssen, 1982). The growth rate is proportional to this curvature and it has an implicit dependence on the roughness length on the wavy water surface. Hence any modification of the vertical wind shear modifies the momentum transfer from wind to waves. This equilibrium is altered for slick-covered surfaces.

[revised manuscript text omitted]

---

## Author Comment (AC2) · 15 Feb 2019

**Revision of the manuscript: os-2018-111**

**Fish oil in a wave tank: a look at the air-water response**

Alvise Benetazzo, Luigi Cavaleri, Hongyu Ma, Shumin Jiang,
Filippo Bergamasco, Wenzheng Jiang, Sheng Chen, and Fangli Qiao

Please find enclosed a detailed list of replies to the Referee 2 comments on the paper in Subject.

We would like to thank the Referee for the comments and suggestions, which have been useful to improve the manuscript.

Sincerely yours,

**Alvise Benetazzo**
Institute of Marine Sciences
National Research Council
Venice, ITALY

**REFEREE #2**

We would like to thank Anonymous Referee #2 for the constructive and helpful comments. Our responses to the comments (in *Italic*) are given below.

REFEREE: *In this article the authors study the impact of fish oil surfactant on the generation of surface gravity wave and on its feedback on the wind profile. To this purpose they use an experimental setup. They shows that when surface wave are inhibited by fish oil, the wind induce surface current is very similar to the one with surface waves. The overall document read wells and the scientific methods and experimental set up seem robust, nevertheless is it difficult from the introduction to understand the real motivation and scientific questioning of the paper which can be seen as a enumeration of different statements and results which would need to be articulated and discussed in regard to a clear scientific questioning. Therefore, although I recognize the quality of the work and the potential of the papers results I recommend to better focus the aim of the works and the implication for ocean atmosphere exchanges before publication of the manuscript.*

AUTHORS: We thank the reviewer for the positive comment of our work and for the suggestion on how better focusing the objectives of our study. Indeed, we recognize that the Introduction and the presentation of the results of the early version of the manuscript could be improved. Therefore, in the revised version of the manuscript we have specifically focused on the research context of our work and on the broad significance of the study.

In this respect, we made clear in the Introduction that we firstly aim at addressing the general question on how a surfactant with high dilatational modulus significantly alters the generation of wind waves and consequently the momentum transfer across the air-water interface. Secondly, we verify for the first time the effectiveness of polar fish oil, often used in the past to smooth angry seas, in suppressing wind waves ("Oil has been known since antiquity to still angry seas, but how it does so remains mysterious" by Cox, C. S., Zhang, X. and Duda, T. F., 2017. Suppressing breakers with polar oil films: Using an epic sea rescue to model wave energy budgets, Geophys. Res. Lett., 44, 1414–1421). Finally, we show the benefits and potentiality of comprehensive laboratory studies with surfactants to explore the interactions between air and water. Moreover, part of the text regarding the Marangoni effect (section 1.1) has been moved to Appendix A ("The Marangoni wave damping effect").

With these and other adjustments of the manuscript (e.g. the description of the interaction between short and long waves has been moved to the Introduction), we focus better on the motivations of our study, including its relevance for air-sea interaction studies. Among the original results we present, for the first time we document that with an effective surfactant the wind-wave field does not grow from the rest condition, but a surface water drift is still present determined by the wind shear only. We have also changed the title of the paper that now reads as "
[revised manuscript text omitted]

Wind blowing over the water surface generates wind waves and drift currents. It is instructive that the physics of this evident truth is still a question of debate. One reason is that the implied physics spans a large range of scales, the various processes interacting among them, and possibly hiding the reasons of that behavior. Nature offers a full panorama of events at all the possible scales. However, it is mainly in the laboratory that we can explore, also with the desired repetitiveness, the details of some, albeit limited in scale, processes. Indeed this approach has provided in time enlightening findings to be then used in the daily operational activities. Already in the '70s and '80s Mitsuyasu, in a series of remarkable papers, provided basic hints into the generative and dissipative processes of wind waves (see, among others, Kusaba and Mitsuyasu, 1984; Mitsuyasu, 1966; Mitsuyasu and Honda, 1982, 1986). Mark Donelan, using data first from a tower in lake Ontario and then in a laboratory wind wave tunnel in Miami, provided basic hints in several aspects of wind wave generation (see e.g. Donelan, 1990). In more recent times, following a very sophisticated and detailed series of experiments, Buckley and Veron (2016) have provided a detailed description of the air flow during wave generation. The specific problem of the trigger of the initial wavelets has been dealt with by Kawai (1979), van Gastel et al. (1985), and more recently by Liberzon and Shemer (2011) and Zavadsky and Shemer (2017).

A problem that (in most of the cases) does not concern the open ocean is how air and sea interact when the water surface is covered by a thin layer of surface active agent. This physical aspect has been early dealt with experimentally in studies by Hühnerfuss et al. (1981) and Mitsuyasu and Honda (1986). The interest is not only on the results of the experiments, but on the physics they reveal and the considerations they allow on the general problem of wind wave generation. Following this logical link we have carried out a series of experiments aimed at, if not solving the whole problem (a daunting task), at least shading new light on some of its aspects. Science proceeds often by negations. New results may not only hint in one direction, but also exclude a solution, in so doing helping focusing along the right path.

In the following we describe what has been done, for each experiment stressing the doubts and the implications. Given that a large part of what was done deals with fish oil on the surface, following subsection 1.1 provides a compact, but sufficient for the purpose, description of the related physics. The general description of the experimental set-up is in section 2, where we also list the general plan of the experiments and the finally available data. The actual technical description of the results of the experiments are in section 3, the whole then discussed in section 4, and summarized and itemized in the final section 5.

**1.1  A little physics on the interaction between oil slick and gravity-capillary waves**

It is well known that the addition of a thin film (from $10^{-9}$ to $10^{-8}$ m, almost mono-molecular) of surfactant (blend of *surface active agent*) to the water surface has a intense effect on the energy of the gravity-capillary waves by altering the surface tension at the water-air interface (Fiscella et al., 1985). Oil has been used for centuries to smooth the sea surface, so much that expressions as "to pour oil on troubled water" have acquired a more general meaning. See later in the paper the impressive example reported by Cox et al. (2017). Crucial in this respect is the type of oil, in particular its polarity. Mineral oils, so often wrongly used during the Second World War, are not effective because their molecules tend to group together in a heap. On the contrary the polar molecules of fish, and partly also vegetable, oils repel each other. Hence, once poured on water, they tend to distribute rapidly on the available surface acquiring a quasi-monomolecular level. Known since ancient time, this effect was first studied in the 19[th] century by the Italian physicist Carlo Marangoni, hence the official name of the process (Marangoni, 1872). In relatively recent times the first report of Marangoni damping of wave spectra came from Cini et al. (1983) who had noted the effect in the polluted (although by mineral oils) water in the gulf of Genoa, Italy. However, clear evidence of Marangoni damping on slick-covered ocean waves was first presented by Ermakov et al. (1985, 1986) during field experiment in the Black Sea.

This resonance type Marangoni damping (to be soon described) can be effective for surface waves in two possible conditions. The first one is in the open ocean, in which an existing wind-forced wave field travels trough a surfactant patch, with the consequent possibility of detection (lack of return signal) by microwave radars (Feindt, 1985). The second one, typical of laboratory experiments, is where the wind blows over a water surface homogeneously covered (since the rest condition) with a surfactant film. The latter is the one we dealt with in the experiments described in this study.

The presence of an extremely thin (practically mono-molecular) layer of oil on the surface strongly affects the air-sea interaction. In this respect, it is generally agreed that, in clean water, the growth of the first detectable ripples on the water surface is rather well explained by the effect of air turbulence advected by the wind (Phillips, 1957). That process is quickly overtaken as the waves grow by the feedback caused by the wave-induced pressure oscillations in the air, as soon as the airflow vertical profile is modified by waves. Miles (1957) proposed a wave growth mechanism that accounts for this change. This theory was extended and later applied by Janssen (1991) to wave forecasting. Its validity is questioned for very short waves whose phase speed is as slow as the air friction velocity (Miles, 1993). According to the shear-flow model by Miles, waves with phase speed $c$ grow when the curvature in the vertical wind profile, at the height (called critical height) where the wind speeds equals $c$, is negative. As a result, the wind profile changes because of the continuous transfer of energy to the waves (Janssen, 1982). The growth rate is proportional to this curvature and it has an implicit dependence on the roughness length on the wavy water surface. Hence any modification of the vertical wind shear modifies the momentum transfer from wind to waves. This equilibrium is altered for slick-covered surfaces.

[revised manuscript text omitted]

---

## Referee Report (RR1)

**Review of 'Analysis of the effect of fish oil on wind waves and implications for air-water interaction studies', Benetazzo et. al.**

I think this is a very nice manuscript that presents refined measurements on complex problematics. I personally enjoyed reading the manuscript and I think that, while the subject has been studied previously in the literature, the experimental evidences presented here are numerous and of high enough quality to definitely improve the current state of the literature on the topic.

I have a couple of relatively minor comments. I think that they may help still improve the quality of the final paper as well as its potential reach and audience, while requiring relatively little work, so I would like the authors to consider including them in their final manuscript.

- 1: Regarding the reach and novelty of the paper, I think that it is already satisfactory as is, but that a bit more can be done. I am thinking more specifically about 'cross-field' applications of the measurements you perform.

More specifically, there are a few more fields where applications of your findings could be relevant:

– a) The detection and tracking of oil spills. While you talk here about fish oils, and you mention that they present some differences compared with mineral oils, I think that in a modern context there is more interest in the effect of mineral oil leaks on the environment than fish oil. Could you discuss about this in the introduction, and put into context for example oil spill detections based on capillary waves damping? See for example:

'Damping of gravity-capillary waves in the presence of oil slicks according to data from laboratory and numerical experiments', Ermakov et. al. 2012.

'Oil Spill Remote Sensing: A Review', Fingas and Brown.

'Drift and deformation of oil slicks due to surface waves', Christensen and Terrile, 2009.

– b) Another field receiving much attention recently is the interaction between waves and sea ice. This is obviously a 'hot' topic currently due to the current global climate trends. There also, elastic forces appear at the surface of the water masses, but this time due to the presence of ice. This field of research would be worth to mention in your introduction and / or conclusion, as it presents many analogies with your present study and making the waves in ice community aware of your findings will definitely be of interest. The parallels between those fields are visible in several works, for example:

'Transient and steady drift currents in waves damped by surfactants', Christensen 2005.

'Measurements of wave damping by a grease ice slick in Svalbard using off-the-shelf sensors and open source electronics', Rabault et. al. 2017.

'The attenuation of monochromatic surface waves due to the presence of an inextensible cover', Sutherland et. al., 2017.

- 2: p 5 l 21: can you give us an idea of which parameters you use in your Welch transformation, and why you chose them so? I ask about this because the choice of those parameters may participate in influencing the shape of your spectra later in the manuscript.

- 3: p 14 Fig. 4: Are you sure that the W06-O signal is well resolved? In particular, you need a signal-to-noise ration high enough to trust the graph shown on the right panel. I ask about this because the red curve has a scale much collapsed compared with the blue one, and therefore I cannot visually check how well you can resolve it. Consider adding a small sentence discussing the accuracy of you measurements in the methodology section, and discussing it in regards to your measurements here.

- 4: p 15 Fig. 5: Why do you provide the ^-4 and ^-5 slopes? Please make it clearer in the text what you want to show with them. Also, if you still want to keep them, please add a sentence or two discussing how compelling evidence they provide. Your spectra display a power law behavior for ranges of frequency that span typically slightly less than a decade, which is arguably quite little.

- 5: p 17 Fig. 7: Please add error bars. For some help about error bars, you may consider reading for example: https://www.ittc.info/media/8099/75-02-07-014.pdf , 'Confidence Intervals for Significant Wave Height and Modal Period'. Similarly, Sutherland et. al. previously cited also show some ways of obtaining error bars on this kind of statistical quantities.

- 6: p 17 l. 20: sec → s

- 7: p 18 Fig. 8: Do the spikes in your figure correspond to noise or waves? Here also an estimate of the Signal To Noise Ratio would help.

- 8: p 23 Fig. 13: same comment about confidence intervals as for Fig. 7.

---

## Author Response (AR2)

**Revision of the manuscript: os-2018-111**

**Analysis of the effect of fish oil on wind waves and implications for air-water interaction studies**

Alvise Benetazzo, Luigi Cavaleri, Hongyu Ma, Shumin Jiang,

Filippo Bergamasco, Wenzheng Jiang, Sheng Chen, and Fangli Qiao

Dear Editor,

Please find enclosed a detailed list of replies to comments on the paper in Subject. We would like to thank for comments and suggestions, which have been useful to improve the manuscript.

Sincerely yours,

**Alvise Benetazzo**

Institute of Marine Sciences

National Research Council

Venice, ITALY

Thank-you for your extensively revised manuscript. I intend to send it back to the reviewers in due course. However, first I think you still need to improve your response to their main questions: what is your aim regarding the gap(s) in existing knowledge or understanding that you intend to fill? What is new about your results.

- We thank the editor for his valuable help in improving our manuscript. We have replied below to his specific comments and, as suggested, modified the response to reviewers, as well as the text of the manuscript. We believe that it is now clear which are the motivations of our study. On the one hand, we were interested in quantifying (for the first time by means of laboratory experiments) the effectiveness of fish oil in impeding the short wind wave generation and growth. Indeed, fish oil effect on the growth of the wave field has been known qualitatively since ancient times, but its effects have never been accurately verified with laboratory experiments. On the other hand, we aimed at using the experiments with surfactant to isolate relevant mechanisms for the air-sea interaction; in this respect, our interest was on the pure frictional wind stress and the long-wave growth. We note that the surfactants that were tested in previous studies do not allow these characterizations since short wind waves remained only partially suppressed.

There are still some places where I think you have unnecessary text which tends to obscure your "message". Please see details below.

**Abstract**.

Page 1 lines 24, 26, 28-29. These statements all refer to similar results with and without oil. So what difference does the oil make?

- The editor is correctly pointing out that for some relevant processes of the air-sea interaction the oil presence makes a very small difference. This fact has never been verified before and it is an original result of our study. These processes occurred to be the irregular long wave growth under the action of the wind and the water surface current generation. The former is mainly due to the suppression of the dominant short waves while interacting with the long waves. This implies that the oil is less effective on short waves with respect to the case where only wind waves are present in the tank. Hence, the wind shear and the momentum transfer to the water mass in both cases are mostly dominated by the long waves. As far as the surface water current generation is concerned, by suppressing the short wave field with surfactants we isolated the pure frictional component of the wind stress, which is used for the generation of the current. In this respect, it is a genuine result of our study that with and without oil the surface current appears to have similar intensity. All these aspects and their relevance, we believe, are described and discussed in our study.

**Introduction**

Page 2 line 8. "in a heap" is not good English and very unclear in meaning.

- We accept the suggestion. In the revised manuscript "in a heap" has been removed.

**Page 3**

Line 2. "since" -> "after" or "at"

- "Since" has been replaced with "from".

Line 16. You state "The growth rate is proportional to this curvature . ." as a fact so a reference is needed.

- We have added the reference to Janssen (1991).

Line 17. ". . Hence it is expected that any modification . ."

- Thanks. The text has been corrected accordingly.

Line 19. Delete "In this respect" (unnecessary)

- Thanks. The text has been corrected accordingly.

**Page 4** line 10. You have said a lot about what previous work has been done. But what is the gap? This is the place to say what gap in knowledge or understanding you are addressing. "complements previous . . investigations" is too vague. "open new perspectives . ." is too abstract. What are your specific objectives?

- We thank the editor for the suggestion. We have thus reshaped the sentence to make clear our objectives. The sentence now reads as "Given these previous experiences, the present study is motivated, on the one hand, to quantify for the first time the effect of fish oil on the generation of the gravity-capillary wind-wave field. We worked with waves generated in a wind-flume using clean water and water with surfactant on the surface. Irregular paddle waves coexisting with wind waves were also tested to investigate the mutual interaction between short waves, long waves, and airflow. On the other hand, using the suppressed wind waves with surfactants, we aimed at using the experiments to disentangle and analyze relevant mechanisms of the air-sea interaction; our interest was on the pure frictional wind stress (which, in clean water, is not separable from the wave stress), the related water surface drift, and the growth of irregular long waves under the action of the wind (by largely canceling the short-wave roughness).".

**Section 3.1.1.**

Page 12 line 16. "Albeit" -> "Despite".

- Thanks. The text has been corrected accordingly.

**Page 13**

Line 2. I do not like "very close to zero" – no criterion for "close". Best to give a value.

- We have provided the order of magnitude " $< 10^{-6}$ m".

Lines 3-4. "supports the idea presented in Mitsuyasu and Honda (1986) and discussed by Mitsuyasu (2015)". This does not help you to claim that your work is novel.

- We really thank the editor to point out this sentence giving us the opportunity to reconsider the Mitsuyasu's results. Indeed, what Mitsuyasu had shown is that the water surface for clean water and for water with surfactants shows aerodynamically similar properties below about U10 = 8 m/s. However, in those experiments the surfactant was not able to fully suppress the wind wave field, hence small oscillations were still present on the water surface. Our experiments with fish oil, on the contrary, showed that this surfactant cancels the wind-wave generation at all scales. This is the reason why the water surface in our case is felt by the airflow as being aerodynamically smooth. There are hence differences between our results and Mitsuyasu's that we did not appreciate before. For this reason, the sentence has been re-formulated and now reads as "This result does not contradict, but complements, those of Mitsuyasu and Honda (1986) and Mitsuyasu (2015), who observed that for low wind speeds (few meters per second) and short fetches the water surface aerodynamic properties are similar in clean water and in water with surfactant. Indeed, in those studies the surfactant only partially suppressed the wind-wave field, while, on the contrary, the use of fish oil in our experiments cancels the wind-wave generation process such that the water surface is felt as smooth by the airflow"

**Section 3.1.2**

Page 13 line 10. Better ". . the process involves more than a decrease . ."

- Thanks. The text has been corrected accordingly.

**Page 14**

Line 10. Better ". . wavenumbers are shifted by the surface current uw0 ="

- Thanks. The text has been corrected accordingly.

Lines 12-13. Omit "As physically sound," (unnecessary). Then better "From now on we only deal with . ."

- Thanks. The text has been corrected accordingly.

**Page 17 lines 15-16**. Omit "Incidentally" and "one point we had deliberately avoided while discussing the spectra in Figure 5, i.e.". (unnecessary). Hence ". . forces. This explains why the highest energy . ."

- Thanks. The text has been corrected accordingly.

**Section 3.2**

**Page 19.**

Lines 5-6. Omit "Within the general purpose of a better understanding of the related air-sea interactions," (unnecessary).

- Thanks. The text has been corrected accordingly.

**Lines 9-10.** Omit "Of course a 6 m/s value would have allowed a more direct comparison with the results obtained without paddle waves (previous section). At the same time," (unnecessary? You do have some comparison later).

- Following the editor's suggestion we have reshaped the sentence.

**Lines 11-13.** Omit "Originally we had planned two full sets of parallel experiments. However, as mentioned in section 2, the strict condition of analyzing only good quality data left us with what is listed in Table 1." (unnecessary; you did mention this in section 2).

- Thanks. The text has been corrected accordingly.

**Lines 19-20**. "by a wind blowing at the reference speed of 8 m/s (experiment W08-P)." -> "by the wind." (repetition).

- Thanks. The text has been corrected accordingly.

**Page 20**

Lines 8-9. Omit "Ignoring for the time being the oily results (experiment W08-P-O)," (unnecessary).

- Thanks. The text has been corrected accordingly.

**Lines 9-10**. Omit "(with a JONSWAP spectrum)" (repetition).

- Thanks. The text has been corrected accordingly.

**Page 21 lines 3-4**. Better "damping (D) of wind wave energy (see for comparison Figure 6) is two or three orders of magnitude smaller . ."

- Thanks. The text has been corrected accordingly.

**Page 23 line 6**. ". . similar to the result found by Mitsuyasu and Honda (1986) . .". This does not help you to claim that your work is novel.

- Of course we do not want to claim the all our results are fully novel. Here the difference between Mitsuyasu and Honda (1986) results and ours is that in our experiments we used irregular waves instead of monochromatic ones. This should lead to an easier generalization towards real oceanic waves. This point has been made clear in the revised version of the manuscript.

**Section 4**

Page 24 lines 17-18. This sentence is too vague – unclear meaning. Be explicit.

- We do agree. The sentence is unclear and thus has been removed from the revised manuscript.

**Page 25**

Lines 2-3. "consistent with the one derived from the wind speed and similar conditions in the literature.". This does not help you to claim that your work is novel.

- What we want to mention here is that using the video data and two original methods (in clean water and in water with oil) we found surface current information consistent with others reported in the literature (in clean water only). We believe that this agreement strengthens the analysis we made. This permits to draw a conclusion about the surface drift in water with oil.

Line 23. Why "However,"? This implies that the following text apparently contradicts the previous text. But I need more explanation to see the contradiction.

- We do agree. "However" is not the right word to introduce the following sentence. We have replaced it with "In this regard".

**Section 5**

Page 26 line 20. "in reference to" -> "compared with"?

- Thanks. The text has been corrected accordingly.

**Page 27**

Line 11. "As mentioned in the text, we have barely touched the surface of the subject." This does not help you to claim that your work is novel.

- The sentence has been changed and now reads as "With our study we have shown that wave generation and dissipation, and more general atmosphere and sea dynamic interaction, are still to be fully explored".

Line 13. Better word than "digging"? E.g. "exploring", "studying"?

- In the revised manuscript we have used "studying".

**Appendix A**

Page 28 line 3. "modules" -> "modulus"

- Thanks. The text has been corrected accordingly.

**Revision of the manuscript: os-2018-111**

**Fish oil in a wave tank: a look at the air-water response**

Alvise Benetazzo, Luigi Cavaleri, Hongyu Ma, Shumin Jiang,
Filippo Bergamasco, Wenzheng Jiang, Sheng Chen, and Fangli Qiao

Dear Sirs,
Please find enclosed a detailed list of replies to the Referees comments on the paper in Subject.

We would like to thank the Referees for their comments and suggestions, which have been useful to improve the manuscript.

Sincerely yours,

**Alvise Benetazzo**
Institute of Marine Sciences
National Research Council
Venice, ITALY

**REFEREE #1**

We would like to thank Anonymous Referee #1 for the constructive and helpful comments. Our responses to the comments (in blue) are given below.

We thank the reviewer for pointing out that the experiments carried out for this study are of good quality and that they can provide useful results. However, to account for the criticisms raised by the reviewer, the manuscript has been partially reshaped and rewritten. Below, our replies to the specific comments are provided.

REFEREE: The manuscript "Fish oil in a wave tank: a look at the air-water response" is not suitable for publication in the present form because the text lacks clarity and structure to such a degree that I find it impossible to evaluate the quality of the underlying scientific work. It is also not clear how the work relates to previous studies of the past 30 years, and what any potential new findings are. The first part of the introduction (line 1-30) do not communicate relevant information about the scientific background or this study but instead seem more like a philosophical discussion about some scientific methods, for instance, line in 18-23: "The interest is not only on the results of the experiments, but on the physics they reveal and the considerations they allow on the general problem of wind wave generation. Following this logical link we have carried out a series of experiments aimed at, if not solving the whole problem (a daunting task), at least shading new light on some of its aspects. Science proceeds often by negations. New results may not only hint in one direction, but also exclude a solution, in so doing helping focusing along the right path." Section 1.1 then proceeds with text on the physical background, but does not prepare the reader for the work that follows. For instance, large parts of the paper seem to be about Marangoni damping. In line 17 in section 1.1, the author write "This resonance- type Marangoni damping (to be soon described)..." , but it is never described in the following text. The next time Marangoni damping is mentioned, the authors already assume that the reader is familiar with it. The ongoing discussion in the introduction jumps form topic to topic without a continuous thread the reader could follow. The authors do not present what the motivation for this study is (is the goal to "still angry waters"?), nor do they present a knowledge gap that they want to address or a goal for this study.

- We thank the reviewer for this keen comment that helped to improve our paper. Indeed, we agree that in the original manuscript the description of the purpose and originality of our research could be improved. Following the reviewer's suggestions, in the revised paper we have hence highlighted in the "Introduction" the motivations of our study. On the one hand, our research is motivated to quantify for the first time by means of laboratory experiments the effect of fish oil on the generation and growth of the gravity-capillary wave field, and on the wind shear. Indeed, fish oil effect on the growth of the wave field has been known qualitatively since ancient times, but its effects have never been accurately verified with laboratory experiments. On the other hand, we aimed at using the experiments with surfactant to disentangle the relevant mechanisms involved in the air-sea interaction. We had two main interests. The first was one was the pure frictional wind stress, able to generate a surface current. Indeed, for the first time we document that, despite with oil the wind-wave field does not grow from the rest condition, a surface water drift is

present driven by the frictional stress. Then we analyzed the irregular long wave growth with almost suppressed short waves. To make the motivations clearer, as suggested, we have hence reshaped abstract, introduction (most of the section 1.1 has been moved to Appendix A "The Marangoni wave damping effect"), text (e.g. the description of the interaction between short and long waves has been moved to the Introduction), and modified the title that now reads as "Analysis of the effect of fish oil on wind waves and implications for air-water interaction studies".

REFEREE: Section 3 (the results) continues to provide introductory material, instead of focusing on presenting the results. I doubt that all this information is needed to discuss their experiments and results. Background knowledge, results, and discussions are intertwined throughout the entire result section which makes it very difficult to read.

- In Section 3, material pertaining to the results and their interpretation is presented. Having clarified in the Introduction the purposes of the present study, this section is better framed in the flow of the manuscript. Some modifications have been however made to Section 3, moving, as suggested, part of the material to Section 2 describing the experiments.

REFEREE: The discussion section does not establish a link to previous works, e.g. experimental studies on oil in a wave tank or similar. The authors claim that they provide a now look at air-sea interaction, but it is not clear to me what the new findings are. In the conclusion section, one finally learns what the aim of the study is (to evaluate the influence of the oil film on the frequency spectrum and growth of waves), but I doubt that these are new finding and they should be discussed in context with recent studies. Beside the quality of how the work is presented, I am also in doubt if new knowledge is provided in this study. Most of the presented references on water-oil-wave interaction are more than 30 years old, and the conclusion that an oil film dampens wind-waves but not swell is considered general knowledge. It should be made clear what new findings is, and what confirmations of previous experimental or theoretical results are.

- To the best of our knowledge, the oil effect in smoothing angry seas is not fully understood. While the Marangoni damping is well described and its modification on the wave spectrum can be predicted (see e.g. Alpers and Hühnerfuss, 1989), the whole interplay between wind shear stress and surface wave source functions is still and open question (Cox et al., 2017. "Oil has been known since antiquity to still angry seas, but how it does so remains mysterious" by Cox, C. S., Zhang, X. and Duda, T. F., 2017. Suppressing breakers with polar oil films: Using an epic sea rescue to model wave energy budgets, Geophys. Res. Lett., 44, 1414–1421). This fact largely stimulated our research. Moreover, the references we mention trace back to 30 years because since then no thorough investigations have been made on the suppression of wind waves using surfactants (neither in open sea nor in a laboratory). Our study was thus aimed at quantifying for the first time the fish oil effect on the generation of wind waves in a tank, but also exploiting these peculiar experiments to investigate processes involved in the air-sea interaction. We mention the original results we obtained on the surface current that was observed and measured in absence of waves (almost

fully damped by the oil slick) and the investigation of the interplay between irregular long and short waves in receiving wind input in clean water and with suppressed short waves by oil.

REFEREE: It seems to me that the experiments carried out for this study are of good quality, and that they can provide useful results. However, the paper should be entirely re-written, focusing on the experiments and results instead of philosophical questions and needles history details.

- We thank the reviewer for pointing out the overall merit of our study.

**REFEREE #2**

We would like to thank Anonymous Referee #2 for the constructive and helpful comments. Our responses to the comments (in blue) are given below.

REFEREE: In this article the authors study the impact of fish oil surfactant on the generation of surface gravity wave and on its feedback on the wind profile. To this purpose they use an experimental setup. They shows that when surface wave are inhibited by fish oil, the wind induce surface current is very similar to the one with surface waves. The overall document read wells and the scientific methods and experimental set up seem robust, nevertheless is it difficult from the introduction to understand the real motivation and scientific questioning of the paper which can be seen as a enumeration of different statements and results which would need to be articulated and discussed in regard to a clear scientific questioning. Therefore, although I recognize the quality of the work and the potential of the papers results I recommend to better focus the aim of the works and the implication for ocean atmosphere exchanges before publication of the manuscript.

- We thank the reviewer for the positive comment of our work and for the suggestion on how better focusing the objectives of our study. Indeed, we recognize that the Introduction and the presentation of the results of the early version of the manuscript could be improved. Therefore, in the revised version of the manuscript we have specifically focused on the research context of our work and on the broad significance of the study. In this respect, we made clear in the Introduction that the present study is motivated, on the one hand, to quantify for the first time by means of laboratory experiments the effect of fish oil on the generation and growth of the gravity-capillary wave field, and on the wind shear. Indeed, fish oil effect on the growth of the wave field has been known qualitatively since ancient times, but its effects have never been accurately verified with laboratory experiments. On the other hand, we aimed at using the experiments with surfactant to disentangle the relevant mechanisms involved in the air-sea interaction; our interest was on the pure frictional wind stress, able to generate a surface current (for the first time we

document that, despite with oil the wind-wave field does not grow from the rest condition, a surface water drift is present driven by the frictional stress), and the irregular long wave growth with suppressed short waves. In the reorganization of the manuscript, part of the text regarding the Marangoni effect (section 1.1) has been moved to Appendix A ("The Marangoni wave damping effect"). With these and other adjustments of the manuscript (e.g. the description of the interaction between short and long waves has been moved to the Introduction), we focus better on the motivations of our study, including its relevance for air-sea interaction studies. We have also changed the title of the paper that now reads as "
[revised manuscript text omitted]

Wind blowing over the water surface generates wind waves and drift currents. It is instructive that the physics of this evident truth is still a question of debate. One reason is that the implied physics spans a large range of scales, the various processes interacting among them, and possibly hiding the reasons of that behavior. Nature offers a full panorama of events at all the possible scales. However, it is mainly in the laboratory that we can explore, also with the desired repetitiveness, the details of some, albeit limited in scale, processes. Indeed this approach has provided in time enlightening findings to be then used in the daily operational activities. Already in the '70s and '80s Mitsuyasu, in a series of remarkable papers, provided basic hints into the generative and dissipative processes of wind waves (see, among others, Kusaba and Mitsuyasu, 1984; Mitsuyasu, 1966; Mitsuyasu and Honda, 1982, 1986). Mark Donelan, using data first from a tower in lake Ontario and then in a laboratory wind wave tunnel in Miami, provided basic hints in several aspects of wind wave generation (see e.g. Donelan, 1990). In more recent times, following a very sophisticated and detailed series of experiments, Buckley and Veron (2016) have provided a detailed description of the air flow during wave generation. The specific problem of the trigger of the initial wavelets has been dealt with by Kawai (1979), van Gastel et al. (1985), and more recently by Liberzon and Shemer (2011) and Zavadsky and Shemer (2017).

A problem that (in most of the cases) does not concern the open ocean is how air and sea interact when the water surface is covered by a thin layer of surface active agent. This physical aspect has been early dealt with experimentally in studies by Hühnerfuss et al. (1981) and Mitsuyasu and Honda (1986). The interest is not only on the results of the experiments, but on the physics they reveal and the considerations they allow on the general problem of wind wave generation. Following this logical link we have carried out a series of experiments aimed at, if not solving the whole problem (a daunting task), at least shading new light on some of its aspects. Science proceeds often by negations. New results may not only hint in one direction, but also exclude a solution, in so doing helping focusing along the right path.

In the following we describe what has been done, for each experiment stressing the doubts and the implications. Given that a large part of what was done deals with fish oil on the surface, following subsection 1.1

provides a compact, but sufficient for the purpose, description of the related physics. The general description of the experimental set-up is in section 2, where we also list the general plan of the experiments and the finally available data. The actual technical description of the results of the experiments are in section 3, the whole then discussed in section 4, and summarized and itemized in the final section 5.

**1.1 A little physics on the interaction between oil slick and gravity-capillary waves**

It is well known that the addition of a thin film (from $10^{-9}$ to $10^{-8}$ m, almost mono-molecular) of surfactant (blend of *surface active agent*) to the water surface has a intense effect on the energy of the gravity-capillary waves by altering the surface tension at the water-air interface (Fiscella et al., 1985). Oil has been used for centuries to smooth the sea surface, so much that expressions as "to pour oil on troubled water" have acquired a more general meaning. See later in the paper the impressive example reported by Cox et al. (2017). Crucial in this respect is the type of oil, in particular its polarity. Mineral oils, so often wrongly used during the Second World War, are not effective because their molecules tend to group together in a heap. On the contrary the polar molecules of fish, and partly also vegetable, oils repel each other. Hence, once poured on water, they tend to distribute rapidly on the available surface acquiring a quasi-monomolecular level. Known since ancient time, this effect was first studied in the 19$^{th}$ century by the Italian physicist Carlo Marangoni, hence the official name of the process (Marangoni, 1872). In relatively recent times the first report of Marangoni damping of wave spectra came from Cini et al. (1983) who had noted the effect in the polluted (although by mineral oils) water in the gulf of Genoa, Italy. However, clear evidence of Marangoni damping on slick-covered ocean waves was first presented by Ermakov et al. (1985, 1986) during field experiment in the Black Sea.

This resonance type Marangoni damping (to be soon described) can be effective for surface waves in two possible conditions. The first one is in the open ocean, in which an existing wind-forced wave field travels trough a surfactant patch, with the consequent possibility of detection (lack of return signal) by microwave radars (Feindt, 1985). The second one, typical of laboratory experiments, is where the wind blows over a water surface homogeneously covered (since the rest condition) with a surfactant film. The latter is the one we dealt with in the experiments described in this study.

The presence of an extremely thin (practically mono-molecular) layer of oil on the surface strongly affects the air-sea interaction. In this respect, it is generally agreed that, in clean water, the growth of the first detectable ripples on the water surface is rather well explained by the effect of air turbulence advected by the wind (Phillips,

1957). That process is quickly overtaken as the waves grow by the feedback caused by the wave-induced pressure oscillations in the air, as soon as the airflow vertical profile is modified by waves. Miles (1957) proposed a wave growth mechanism that accounts for this change. This theory was extended and later applied by Janssen (1991) to wave forecasting. Its validity is questioned for very short waves whose phase speed is as slow as the air friction velocity (Miles, 1993). According to the shear-flow model by Miles, waves with phase speed $c$ grow when the curvature in the vertical wind profile, at the height (called critical height) where the wind speeds equals $c$, is negative. As a result, the wind profile changes because of the continuous transfer of energy to the waves (Janssen, 1982). The growth rate is proportional to this curvature and it has an implicit dependence on the roughness length on the wavy water surface. Hence any modification of the vertical wind shear modifies the momentum transfer from wind to waves. This equilibrium is altered for slick-covered surfaces.

[revised manuscript text omitted]

---

## Author Response (AR3)

**Revision of the manuscript: os-2018-111**

**Analysis of the effect of fish oil on wind waves and implications for air-water interaction studies**

Alvise Benetazzo, Luigi Cavaleri, Hongyu Ma, Shumin Jiang,
Filippo Bergamasco, Wenzheng Jiang, Sheng Chen, and Fangli Qiao

Dear Sirs,
Please find enclosed a detailed list of replies to the Referee comments on the paper in Subject.

We would like to thank the Referee for her/his comments and suggestions, which have been useful to improve the manuscript.

Sincerely yours,

**Alvise Benetazzo**
Institute of Marine Sciences
National Research Council
Venice, ITALY

**1.1   REFEREE #1**

*We would like to thank the Referee for the constructive and helpful comments. Our responses (in blue) to the comments are given below.*

REFEREE: I think this is a very nice manuscript that presents refined measurements on complex problematics. I personally enjoyed reading the manuscript and I think that, while the subject has been studied previously in the literature, the experimental evidences presented here are numerous and of high enough quality to definitely improve the current state of the literature on the topic.

I have a couple of relatively minor comments. I think that they may help still improve the quality of the final paper as well as its potential reach and audience, while requiring relatively little work, so I would like the authors to consider including them in their final manuscript.

- 1: Regarding the reach and novelty of the paper, I think that it is already satisfactory as is, but that a bit more can be done. I am thinking more specifically about 'cross-field' applications of the measurements you perform. More specifically, there are a few more fields where applications of your findings could be relevant:

– a) The detection and tracking of oil spills. While you talk here about fish oils, and you mention that they present some differences compared with mineral oils, I think that in a modern context there is more interest in the effect of mineral oil leaks on the environment than fish oil. Could you discuss about this in the introduction, and put into context for example oil spill detections based on capillary waves damping? See for example:

- 'Damping of gravity-capillary waves in the presence of oil slicks according to data from laboratory and numerical experiments', Ermakov et. al. 2012.
- 'Oil Spill Remote Sensing: A Review', Fingas and Brown.
- 'Drift and deformation of oil slicks due to surface waves', Christensen and Terrile, 2009.

AUTHORS: We thank the Referee for this useful comment and for the suggested references. In this respect, we have mentioned in the "Introduction" the connection between our study and the "remote detection of oil spills". The two relevant studies by Fingas and Brown (2017) and Christensen and Terrile (2009) have been cited accordingly.

– b) Another field receiving much attention recently is the interaction between waves and sea ice. This is obviously a 'hot' topic currently due to the current global climate trends. There also, elastic forces appear at the surface of the water masses, but this time due to the presence of ice. This field of research would be worth to mention in your introduction and / or conclusion, as it presents many analogies with your present study and making the waves in ice community aware of your findings will definitely be of interest. The parallels between those fields are visible in several works, for example:

- 'Transient and steady drift currents in waves damped by surfactants', Christensen 2005.
- 'Measurements of wave damping by a grease ice slick in Svalbard using off-the-shelf sensors and open source electronics', Rabault et. al. 2017.
- 'The attenuation of monochromatic surface waves due to the presence of an inextensible cover', Sutherland et. al., 2017.

We thank the referee for highlighting the parallel between those two relevant fields. In this respect, we have mentioned in the "Introduction" the similarity of the problem we investigated with the "wave attenuation under an ice cover". The two studies by Weber (1987) and Christensen (2005) have been added to the reference lists.

- 2: p 5 l 21: can you give us an idea of which parameters you use in your Welch transformation, and why you chose them so? I ask about this because the choice of those parameters may participate in influencing the shape of your spectra later in the manuscript.

For the spectral analysis we have used the Welch method adopting eight segments with 50% overlap. Each segment was windowed with a Hamming window. We made sensitivity tests by changing the overlap and the window's type, but we noted very minor differences in the spectrum shape. Following the referee's suggestion, in the revised version of the manuscript we have specified the Welch method parameter: "eight segments of equal length, 50% overlap, Hamming window".

- 3: p 14 Fig. 4: Are you sure that the W06-O signal is well resolved? In particular, you need a signal-to-noise ration high enough to trust the graph shown on the right panel. I ask about this because the red curve has a scale much collapsed compared with the blue one, and therefore I cannot visually check how well you can resolve it.

Consider adding a small sentence discussing the accuracy of you measurements in the methodology section, and discussing it in regards to your measurements here.

We thank the Referee for pointing out this important aspect. The calibrations we carried out before the Experiments showed a linear response of the gauges, whose accuracy was found to be not better than 0.3 mm. This provides a signal-to-noise ratio sufficient for all our analyses, being the standard deviation of the surface elevation during the test W06-O at least twice as large as the accuracy, and the maximum elevation about 2 mm. In the revised version of the manuscript (Section 2.1 "Experimental setup") we have included the value of this accuracy as well as reported the linear-type response of the instrument.

- 4: p 15 Fig. 5: Why do you provide the ^-4 and ^-5 slopes? Please make it clearer in the text what you want to show with them. Also, if you still want to keep them, please add a sentence or two discussing how compelling evidence they provide. Your spectra display a power law behavior for ranges of frequency that span typically slightly less than a decade, which is arguably quite little.

We do agree with the referee that for the results presented in our study those slopes are not particular meaningful. Following the Referee's suggestion we have thus decided to remove those lines from Figures 5 and 8 in the revised version of the manuscript

- 5: p 17 Fig. 7: Please add error bars. For some help about error bars, you may consider reading for example: https://www.ittc.info/media/8099/75-02-07-014.pdf , 'Confidence Intervals for Significant Wave Height and Modal Period'. Similarly, Sutherland et. al. previously cited also show some ways of obtaining error bars on this kind of statistical quantities.

We thank for this suggestion that is helpful to improve the paper. As suggested by the Referee and following the example of Sutherland et al. (2917), we have evaluated the predictions intervals of the fit. They are displayed with black lines in the new Figure 7. We have additionally specified in the Figure's caption the root-mean-square error for each fit.

- 6: p 17 l. 20: sec → s

Thanks. Corrected

- 7: p 18 Fig. 8: Do the spikes in your figure correspond to noise or waves? Here also an estimate of the Signal To Noise Ratio would help.

The data displayed do represent waves, even though, we agree, they are not straightly detectable in the Figure. However, we have included in the revised version of the manuscript the expected accuracy for wave probes, hence it is straightforward to estimate the reliability of the plotted data (see surface elevations are in the order of few mm). For the sake of review, we provide below a zoom of the Figure.

[Figure]

- 8: p 23 Fig. 13: same comment about confidence intervals as for Fig. 7.

As for Figure 7, we have added in the new Figure 13 the confidence bounds for the fitted curves.

[revised manuscript text omitted]

Wind blowing over the water surface generates wind waves and drift currents. It is instructive that the physics of this evident truth is still a question of debate. One reason is that the implied physics spans a large range of scales, the various processes interacting among them, and possibly hiding the reasons of that behavior. Nature offers a full panorama of events at all the possible scales. However, it is mainly in the laboratory that we can explore, also with the desired repetitiveness, the details of some, albeit limited in scale, processes. Indeed this approach has provided in time enlightening findings to be then used in the daily operational activities. Already in the '70s and '80s Mitsuyasu, in a series of remarkable papers, provided basic hints into the generative and dissipative processes of wind waves (see, among others, Kusaba and Mitsuyasu, 1984; Mitsuyasu, 1966; Mitsuyasu and Honda, 1982, 1986). Mark Donelan, using data first from a tower in lake Ontario and then in a laboratory wind wave tunnel in Miami, provided basic hints in several aspects of wind wave generation (see e.g. Donelan, 1990). In more recent times, following a very sophisticated and detailed series of experiments, Buckley and Veron (2016) have provided a detailed description of the air flow during wave generation. The specific problem of the trigger of the initial wavelets has been dealt with by Kawai (1979), van Gastel et al. (1985), and more recently by Liberzon and Shemer (2011) and Zavadsky and Shemer (2017).

A problem that (in most of the cases) does not concern the open ocean is how air and sea interact when the water surface is covered by a thin layer of surface active agent. This physical aspect has been early dealt with experimentally in studies by Hühnerfuss et al. (1981) and Mitsuyasu and Honda (1986). The interest is not only on the results of the experiments, but on the physics they reveal and the considerations they allow on the general problem of wind wave generation. Following this logical link we have carried out a series of experiments aimed at, if not solving the whole problem (a daunting task), at least shading new light on some of its aspects. Science proceeds often by negations. New results may not only hint in one direction, but also exclude a solution, in so doing helping focusing along the right path.

In the following we describe what has been done, for each experiment stressing the doubts and the implications. Given that a large part of what was done deals with fish oil on the surface, following subsection 1.1 provides a compact, but sufficient for the purpose, description of the related physics. The general description of the experimental set-up is in section 2, where we also list the general plan of the experiments and the finally available data. The actual technical description of the results of the experiments are in section 3, the whole then discussed in section 4, and summarized and itemized in the final section 5.

**2.1   A little physics on the interaction between oil slick and gravity-capillary waves**

It is well known that the addition of a thin film (from $10^{-9}$ to $10^{-8}$ m, almost mono-molecular) of surfactant (blend of *surface active agent*) to the water surface has a intense effect on the energy of the gravity-capillary waves by altering the surface tension at the water-air interface (Fiscella et al., 1985). Oil has been used for centuries to smooth the sea surface, so much that expressions as "to pour oil on troubled water" have acquired a more general meaning. See later in the paper the impressive example reported by Cox et al. (2017). Crucial in this respect is the type of oil, in particular its polarity. Mineral oils, so often wrongly used during the Second World War, are not effective because their molecules tend to group together in a heap. On the contrary the polar molecules of fish, and partly also vegetable, oils repel each other. Hence, once poured on water, they tend to distribute rapidly on the available surface acquiring a quasi-monomolecular level. Known since ancient time, this effect was first studied in the 19th century by the Italian physicist Carlo Marangoni, hence the official name of the process (Marangoni, 1872). In relatively recent times the first report of Marangoni damping of wave spectra came from Cini et al. (1983) who had noted the effect in the polluted (although by mineral oils) water in the gulf of Genoa, Italy. However, clear evidence of Marangoni damping on slick-covered ocean waves was first presented by Ermakov et al. (1985, 1986) during field experiment in the Black Sea.

This resonance-type Marangoni damping (to be soon described) can be effective for surface waves in two possible conditions. The first one is in the open ocean, in which an existing wind-forced wave field travels trough a surfactant patch, with the consequent possibility of detection (lack of return signal) by microwave radars (Feindt, 1985). The second one, typical of laboratory experiments, is where the wind blows over a water surface homogeneously covered (since the rest condition) with a surfactant film. The latter is the one we dealt with in the experiments described in this study.

The presence of an extremely thin (practically mono-molecular) layer of oil on the surface strongly affects the air-sea interaction. In this respect, it is generally agreed that, in clean water, the growth of the first detectable ripples on the water surface is rather well explained by the effect of air turbulence advected by the wind (Phillips, 1957). That process is quickly overtaken as the waves grow by the feedback caused by the wave-induced pressure oscillations in the air, as soon as the airflow vertical profile is modified by waves. Miles (1957) proposed a wave growth mechanism that accounts for this change. This theory was extended and later applied by Janssen (1991) to wave forecasting. Its validity is questioned for very short waves whose phase speed is as slow as the air friction velocity (Miles, 1993). According to the shear-flow model by Miles, waves with phase speed $c$ grow when the curvature in the vertical wind profile, at the height (called critical height) where the wind speeds equals $c$, is negative. As a result, the wind profile changes because of the continuous transfer of energy to the waves (Janssen, 1982). The growth rate is proportional to this curvature and it has an implicit dependence on the roughness length on the wavy water surface. Hence any modification of the vertical wind shear modifies the momentum transfer from wind to waves. This equilibrium is altered for slick-covered surfaces.

[revised manuscript text omitted]

where $J_\text{ai} = |\text{d}f_\text{a} \,/\, \text{d}f_\text{i}|$ is the Jacobian of the transformation, which in the limit of deep water can be written explicitly as

$$J_\text{ai} = |1 \,+\, 2u_\text{w}\sqrt{gk + \tfrac{T}{\rho}k^3}\Big/\Big(g + 3\tfrac{T}{\rho}k^2\Big)| \qquad (\,6\,)$$

**4    Results**

**4.1    Wind waves without and with oil**

We begin the examination of the change of gravity-capillary wave properties caused by the fish oil film by analyzing the effects on the water elevation $z$ and wind speed profile $U_\text{a}(h)$ during the experiments W06 ($U_\text{r} = 6$ m/s and clean water) and W06-O (the same as W06, but with oil slick). For the latter, the wind wave field attenuation due to the Marangoni forces is readily visible in Figure 2 that shows two pictures of the water surface (taken nearby the 20-m fetch ) without (left panel) and with (right panel)  oil instillation. After the oil film is spread on the water, the surface is largely smooth, with only tiny elevation oscillations (1 mm at most; see Figure 4) and there appears to be no organized wave motion (see also the video available as supplementary material SM1: https://doi.org/10.5281/zenodo.1434262).  It is obvious that the presence of an extremely thin (practically mono-molecular) layer of oil on the surface alters heavily the air-sea interface properties.  We analyze the situation first from the air, then from the water, point of view.

[Figure]

**Figure 2: Two photographs showing the water surface condition at fetch of about 20 m taken without (left panel) and with oil instillation (right panel). On both cases the water surface was forced with a reference wind speed $U_r$ = 6 m/s (blowing from left to right on the pictures). The wave probe G4 is visible on the left-hand side of both pictures. The red arrow in the right panel points to the oscillating flow (vortex shedding) past the probe. See also the video available as supplementary material SM1.**

**4.1.1    Airflow characterization**

We begin evaluating the air-side stress due to the wind drag on the water surface. ~~In this respect, the wind speed vertical profile above the water surface was approximated using the Pitot tube records of the along-channel air speed profile, namely $U_a(h)$. For an aerodynamically rough airflow, the along-channel component of the mean wind speed $\
[revised manuscript text omitted]
 1. As we have seen and will see, the available data suffice for providing a number of remarkable results.

[revised manuscript text omitted]

---

## Author Response (AR4)

**Revision of the manuscript: os-2018-111**

**Analysis of the effect of fish oil on wind waves and implications for air-water interaction studies**

Alvise Benetazzo, Luigi Cavaleri, Hongyu Ma, Shumin Jiang,
Filippo Bergamasco, Wenzheng Jiang, Sheng Chen, and Fangli Qiao

Dear Editor,
Please find below the marked-up manuscript version following the suggested modifications. Moreover, we have added two reference lines in Figure 11 (bottom panels) that were erroneously missing in the previous version.

We would like to thank the editor for comments and suggestions, which have been useful to improve the manuscript.

Sincerely yours,

**Alvise Benetazzo**
Institute of Marine Sciences
National Research Council
Venice, ITALY

[revised manuscript text omitted]